# In vivo quantitative imaging of tumor pH by nanosonophore assisted multispectral photoacoustic imaging

Janggun Jo[1], Chang H. Lee [2], Raoul Kopelman[1,2] & Xueding Wang[1,3]

Changes of physiological pH are correlated with several pathologies, therefore the development of more effective medical pH imaging methods is of paramount importance. Here, we report on an in vivo pH mapping nanotechnology. This subsurface chemical imaging is based on tumor-targeted, pH sensing nanoprobes and multi-wavelength photoacoustic imaging (PAI). The nanotechnology consists of an optical pH indicator, SNARF-5F, 5-(and-6)-Carboxylic Acid, encapsulated into polyacrylamide nanoparticles with surface modification for tumor targeting. Facilitated by multi-wavelength PAI plus a spectral unmixing technique, the accuracy of pH measurement inside the biological environment is not susceptible to the background optical absorption of biomolecules, i.e., hemoglobins. As a result, both the pH levels and the hemodynamic properties across the entire tumor can be quantitatively evaluated with high sensitivity and high spatial resolution in in vivo cancer models. The imaging technology reported here holds the potential for both research on and clinical management of a variety of cancers.

---

[1] Department of Biomedical Engineering, University of Michigan, Ann Arbor, Michigan 48109, USA. [2] Department of Chemistry, University of Michigan, Ann Arbor, Michigan 48109, USA. [3] Department of Radiology, University of Michigan Medical School, Ann Arbor, Michigan 48109, USA. Janggun Jo, and Chang H. Lee contributed equally to this work. Correspondence and requests for materials should be addressed to R.K. (email: kopelman@umich.edu) or to X.W. (email: xdwang@umich.edu)

The regulation of pH is central to the processes of homeostatic control in mammalian tissues. The pH determines the charge state on proteins and macromolecules. Thus pH change can affect various pathologies, such as inflammation, renal disease, ischemia, chronic lung disease and intrauterine disorders[1–3]. In cases of cancer, the tumors are often found in an altered metabolic state, which leads to anomalous extracellular chemical composition, such as acidosis (low pH), hypoxia (low oxygen) and/or hyperkalemia (high potassium)[4–7]. Acidity of the tumor is explained by the Warburg Effect, which describes how lack of nutrients and oxygen converts the tumors' metabolic pathway from aerobic glycolysis into anaerobic glycolysis[8]. A side product of anaerobic glycolysis, lactic acid, is responsible for the acidic tumor microenvironment, the excess lactic acid production surpasses the buffering capacity of the body fluid which thus becomes more acidic[8]. The chemically altered states are believed to cause metastasis, local recurrence and angiogenesis, and are among the causes for failure of cancer treatment[9–11]. Although the importance of pathological acidosis has been established, a simple and generally applicable clinical tool that gives spatially resolved quantitative pH information throughout the tumor is not yet available.

Quantitative in vivo pH imaging methods have been developed mostly for magnetic resonance imaging (MRI)[12–14], positron emission tomography (PET)[15, 16] and optical imaging[17–20]. MRI- and PET-based methods have serious limitations such as high cost and limited access. In addition, MRI cannot be utilized on patients with pacemakers or implants, and PET utilizes ionizing radiation which has safety concerns. Optical spectroscopy based techniques are highly desirable due to many advantages such as low cost, non-invasiveness and ease of use.

Optical pH imaging has been developed for many decades, historically starting with litmus paper. In recent years, numerous near-infrared fluorescent pH indicators have been developed[17–20]. On the basis of the measurement of the pH-dependent emission spectrum, fluorescence microscopy has been heavily used for pH measurements of cells in vitro or of superficial tissues in vivo. However, as a result of the strong optical scattering of biological tissues, traditional optical spectroscopy is not able to achieve satisfactory spatial resolution when the target tissue is beyond the sample surface. This limited spatial resolution in imaging subsurface tumor also inevitably leads to limited accuracy in evaluating the cancer microenvironment, because the functional hallmarks of tumor including the acidosis also show strong heterogeneity.

Combining the merits of light and ultrasound, the emerging non-ionizing and non-invasive photoacoustic imaging (PAI) method holds the potential to overcome these challenges[21–24]. In PAI, the spatial resolution is not limited by the detection of photons which are highly scattered in most soft tissues, but instead by the detection of ultrasound waves which are much less scattered. Thus, the spatial resolution of PAI is comparable to that of ultrasound (US) imaging which, especially when working at high frequencies, is superior in delineating the morphology of subsurface solid tumors. Our group, followed by others, has performed some preliminary pH measurements using PAI[20, 25–29]. A well-known ratiometric pH indicator SNARF-5F, i.e., 5-(and-6)-Carboxylic Acid, has been explored for qualitative pH measurement in vivo. The emission spectrum of SNARF-5F undergoes a pH-dependent wavelength shift, allowing the ratio of the fluorescence intensities at two emission wavelengths to be used for the measurement of pH. Similar to the emission

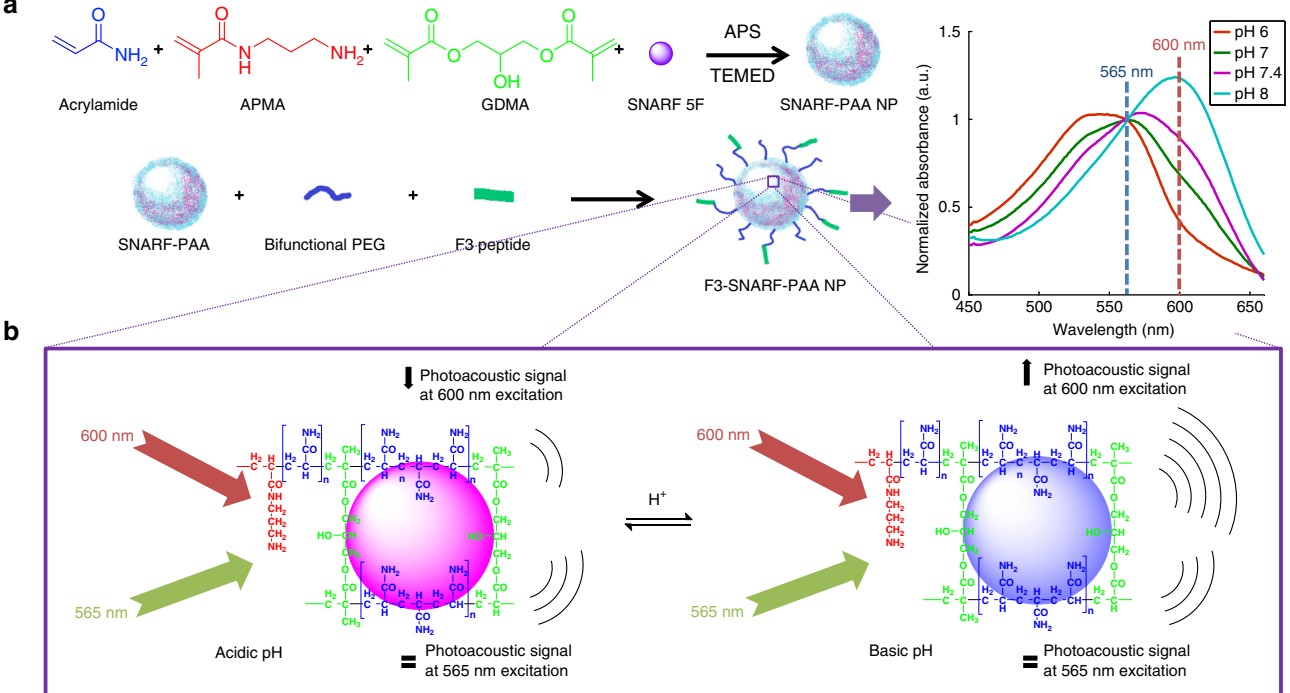

**Fig. 1** SNARF-5F encapsulated polyacrylamide based nanoparticle (NP) synthesis and pH-sensing scheme. **a** Two-step synthesis schematic of SNARF-5F encapsulated acrylamide-based NP. The surface of the SNARF-PAA NP is modified with polyethylene glycol, for immune system avoidance, and with the tumor-homing F3 peptides. The normalized optical absorption spectra at different pH levels indicate the SNARF-PAA NP's capability of pH sensing. The absorption at 565 nm, i.e., the isosbestic point, can be used as an internal reference point. The absorption at 600 nm can be used as a sensing point, which responds to difference in pH level. **b** PA pH-sensing scheme. The PA signal increases linearly with the optical absorption of NPs. Thus, the PA signal amplitude at 600 nm changes as the environmental pH changes; while the PA signal amplitude at 565 nm remains unchanged, acting as an internal reference signal

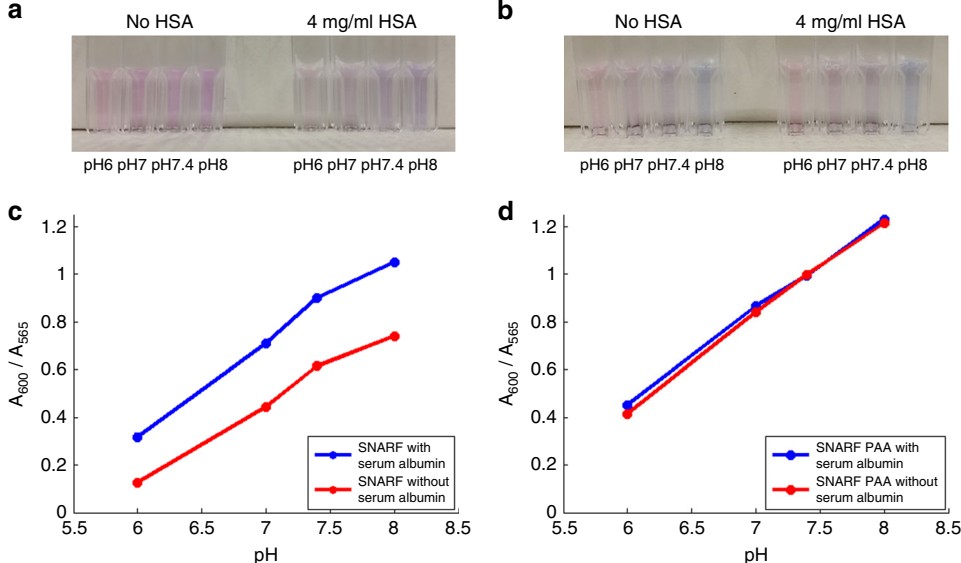

**Fig. 2** The different interactions of free SNARF-5F and SNARF-PAA NP with human serum albumin (HSA). **a** Direct observations of the free SNARF-5F buffered at different pH of 6, 7, 7.4 and 8. The differences in color can be seen for solutions with and without HSA (4 mg ml$^{-1}$). **b** Direct observations of SNARF-PAA NPs buffered at different pH of 6.0, 7.0, 7.4 and 8.0. No obvious difference in color can be seen for solutions with and without HSA (4 mg ml$^{-1}$). **c** pH calibration curves (absorbance at 600 nm divided by absorbance at 565 nm) of free SNARF-5F with (*blue*) and without (*red*) HSA. Significant changes caused by the interaction of SNARF-5F with HSA can be noticed. **d** pH calibration curves of SNARF-PAA NPs with (*blue*) and without (*red*) HSA. No noticeable changes can be noticed, demonstrating the advantages of having the dye encapsulated in the PAA matrix

spectrum, the absorption spectrum of SNARF-5F is also a function of pH. Therefore, following the idea of fluorescent dual-wavelength ratiometric measurement, photoacoustic (PA) absorption dual-wavelength ratiometric measurement of SNARF-5F has been explored, and its capability in detecting as little as a 0.1 pH change has been demonstrated[29]. However, due to the severe background optical absorption in biological tissues (mostly from hemoglobin), none of the previous studies, including ours, has achieved truly quantitative pH measurement of tumor in vivo. When the two forms of hemoglobin also contribute to the PA signals, providing accurate spatial pH information is nearly impossible without careful consideration of the background optical absorption spectrum.

In this study, we developed a method for quantitative pH imaging of tumor in vivo using PAI. To make the pH sensitive nanosonophores, we encapsulated the commercially available optical pH indicator, SNARF-5F, into polyacrylamide nanoparticles (NPs), which we call SNARF-PAA NPs. The NP matrix protects the indicator dye from interacting with large outside molecules, such as albumin, while the surface modification of the NPs makes the pH nano-indicator to be both tumor targeting and immune system avoiding, allowing more efficient and targeted delivery of the pH sensing cargo via intravenous injection[30–33]. As demonstrated by our previous studies, our hydrogel-based standard design of nanoprobes are biocompatible and bio-eliminable[34, 35], benefiting potential translation into clinic. As stated, the NP matrix not only serves as a vehicle for delivery but also prevents SNARF-5F interference with/by body proteins or enzymes. Such interference would affect the optical properties of the pH indicators and thus invalidate the pH calibration. Furthermore, by imaging a tumor with quadruple (i.e., four) laser wavelengths and then performing spectral unmixing, the contributions to the PA signals from the SNARF-PAA NPs and the two forms of hemoglobin (i.e., oxygenated hemoglobin and deoxygenated hemoglobin) can be separated. Thus, this method can provide truly quantitative and spatially resolved pH information in the tumor. At the same time, additional functional bits of information on tumor hemodynamic properties are also mapped, including the spatially distributed total hemoglobin concentration (THb) (i.e., blood volume) and the hemoglobin oxygen saturation (i.e., blood oxygenation).

## Results

**SNARF-5F incorporated polyacrylamide pH-sensing nanoparticles.** As shown in the normalized optical absorption spectra (Fig. 1a), measured by an ultraviolet–visible (UV–VIS) spectrometer, the spectroscopic absorption of SNARF-PAA NPs varies with the pH level. The absorption at 565 nm presents an isosbestic point (i.e., the pH independent point). The isosbestic point can serve as the internal standard without need for a secondary reference probe. The ratio between the intensity at another optical wavelength, such as 600 nm, and that of the isosbestic point correlates with the pH level. The 600 nm point is picked because the optical absorbance of SNARF-PAA NP at this wavelength has a large dynamic range when pH changes from 6 to 8. The selection of this wavelength leads to optimal sensitivity in pH measurement. Our SNARF-PAA NP probe is fully reversible, and can respond to environmental pH changes within a few seconds. The sensing kinetics as well as additional characterization of the SNARF-PAA NP (e.g., size, cytotoxicity, photostability and temperature sensitivity) can be found in Supplementary Figs. 1–5.

In its free form without being protected by NPs, the SNARF-5F can interact with the large proteins in the bloodstream (such as serum albumin). The direct interaction between SNARF-5F and proteins changes the optical properties of the dye and thus affects its pH sensing capability, as demonstrated in Supplementary Fig. 6[29]. The use of NPs can overcome this problem and stabilize the optical properties of the dye in vivo. The PAA matrix, which allows penetration of water and ions but not of protein molecules, essentially protects the dye molecules from the environment, preventing them from directly interacting with the proteins. To demonstrate this improvement, we investigated the interaction of

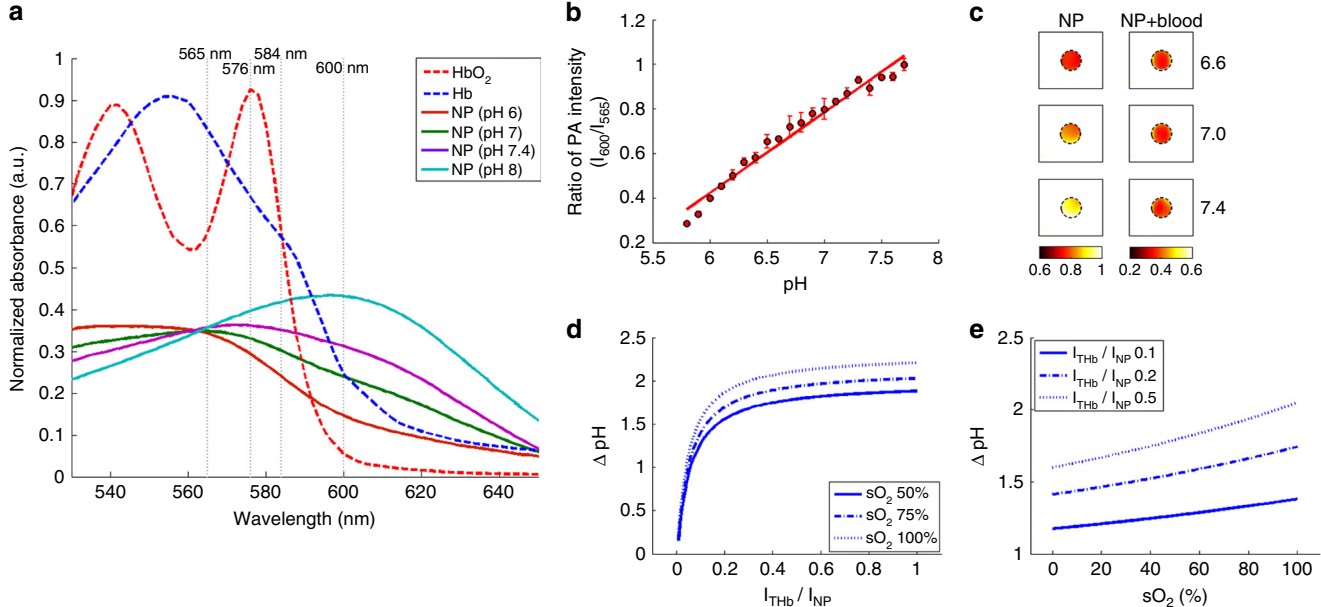

**Fig. 3** Limitation of dual-wavelength ratiometric photoacoustic imaging of pH. **a** Spectroscopic optical absorption of oxygenated hemoglobin (HbO$_2$), deoxygenated hemoglobin (Hb) and SNARF-PAA NPs at different pH levels. **b** Measured PA signal amplitude ratios between the two wavelengths (600 nm/565 nm) as a function of pH (pH 5.8–7.8 with 0.1 pH interval) ($n = 3$, error bars represent standard deviations). By performing a linear fitting of the scattered measurements in the range of pH 5.8–7.8, a calibration line was generated. **c** The PA ratiometric images (600 nm/565 nm) of phantoms containing SNARF-PAA NPs buffered at pH 6.6, 7.0 and 7.4, respectively. The nanoparticle (NP) images are for phantoms with NPs and the NP + blood images are for phantoms with NPs containing blood (1% w/w). The color scale represents the ratio between the PA intensities at the two wavelengths (600 nm/565 nm). **d** Estimated errors (i.e., ΔpH) in dual-wavelength ratiometric PA pH measurement as a function of the ratio between the PA signal amplitudes from total hemoglobin (THb) and SNARF-PAA NPs (i.e., I$_{THb}$/I$_{NP}$). The three curves are for three different hemoglobin oxygen saturation (sO$_2$) levels (50, 75 and 100%). **e** Estimated errors (i.e., ΔpH) in dual-wavelength ratiometric PA pH measurement as a function of the blood sO$_2$. The three curves are for three different ratios of I$_{THb}$/I$_{NP}$ (0.1, 0.2 and 0.5)

SNARF-5F and SNARF-PAA NP with human serum albumin (HSA). The serum albumin is the most abundant protein found in the bloodstream, typically ranging in concentration from 3 to 5 mg ml$^{-1}$[36]. HSA (4 mg ml$^{-1}$) was added to either SNARF-5F free dye or SNARF-PAA NPs in solutions buffered at pH 6.0, pH 7.0, pH 7.4 or pH 8.0. The photographs in Fig. 2a show the color changes of free SNARF-5F after adding of HSA; while no obvious color change can be noticed for the SNARF-PAA NPs after adding HSA, as shown by the photographs in Fig. 2b. The calibration curves (i.e., the optical absorption ratios between 600 and 565 nm at different pH levels) with and without the appearance of HSA were shown in Fig. 2c, d, respectively, for free SNARF-5F and SNARF-PAA NPs. We can clearly see the change in the calibration curve of the free SNARF-5F caused by HSA; while the calibration curve of the SNARF-PAA NPs was unaffected after the adding of HSA. This study suggested that, compared to free SNARF-5F, SNARF-PAA NP is a better choice for in vivo applications where the appearance of proteins cannot be avoided.

**Dual-wavelength ratiometric PAI of pH**. Figure 3a shows the optical absorption spectra of SNARF-PAA NPs at different pH levels. The optical absorption spectra of oxygenated hemoglobin (HbO$_2$) and deoxygenated hemoglobin (Hb), the major two types of hemoglobin in blood, are also presented. Assuming that the pH sensing dye is the only chromophore in the target tissue, ratiometric PA measurements at two laser wavelengths are sufficient for quantifying pH level. Although this assumption has been adopted by many previous studies including ours[29], it cannot stand in most circumstances when the optical absorption from background tissue cannot be ignored. In the visible to nearly

infrared spectral region, the dominant chromophore in most biological tissues is the hemoglobin. When the hemoglobin also contributes to the PA signal, then ratiometric PAI using two laser wavelengths is not sufficient anymore for quantifying pH.

An experiment on phantoms was performed to demonstrate the challenges of ratiometric PA pH imaging based on two wavelengths. Before pH measurement, a calibration line for SNARF-PAA NP was first generated from pH 5.8 to 7.8, as shown in Fig. 3b. This calibration line shows the PA intensity ratios between the two wavelengths (i.e., 600 nm/565 nm) at different pH levels. With this calibration line, any ratiometric PA measurement at the two wavelengths leads to a corresponding pH level. Phantoms were made of porcine gels containing SNARF-PAA NPs, and were buffered at different pH values (i.e., pH 6.6, 7.0, and 7.4). Each phantom was a long cylinder (1.6 mm diameter), and was imaged along the cross-section. In one set of phantoms, no blood was added and, therefore, the SNARF-5F dye was the only optical absorber. In another set of phantoms, whole blood was mixed in each phantom with a concentration of 1% (w/w). Each phantom was imaged at the two laser wavelengths (565 and 600 nm). With the ratiometric measurements, the pixel-by-pixel pH levels in each phantom were calculated. As we can see, for phantoms containing no blood, their pH can be measured well by performing dual-wavelength ratiometric PAI, as shown in Fig. 3c. However, for phantoms containing blood which also contributed to the PA signals, dual-wavelength ratiometric PAI was not able to quantify their pH levels (Fig. 3c).

To understand how much the presence of blood in a target tissue can affect the dual-wavelength ratiometric PAI of pH, a theoretical simulation was conducted. The optical absorption of blood is contributed mainly by the two forms of hemoglobin,

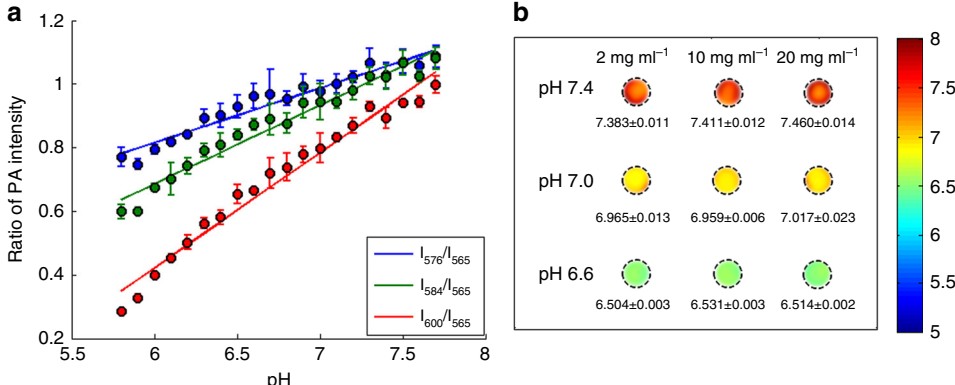

**Fig. 4** Quad-wavelength ratiometric PA pH imaging of phantoms. **a** Measured PA signal amplitude ratios between the three wavelengths and the isosbestic point (i.e., 576 nm/565 nm, 584 nm/565 nm, and 600 nm/565 nm) from pH 5.8–7.7 with 0.1 pH interval ($n = 3$, error bars represent standard deviations). By performing linear fittings of the scattered measurements in the range of pH 5.8–7.8, three calibration lines were generated. **b** Quantitative pH images of phantoms containing different concentrations (2, 10 and 20 mg ml$^{-1}$) of SNARF-PAA NPs buffered at different pH levels (pH 6.6, 7.0 and 7.4). The means and the standard deviations of the pH levels in each PA image were calculated. The measurement accuracy was better than 0.1 pH. The presence of whole blood (1%, w/w) did not affect the quantification of pH levels

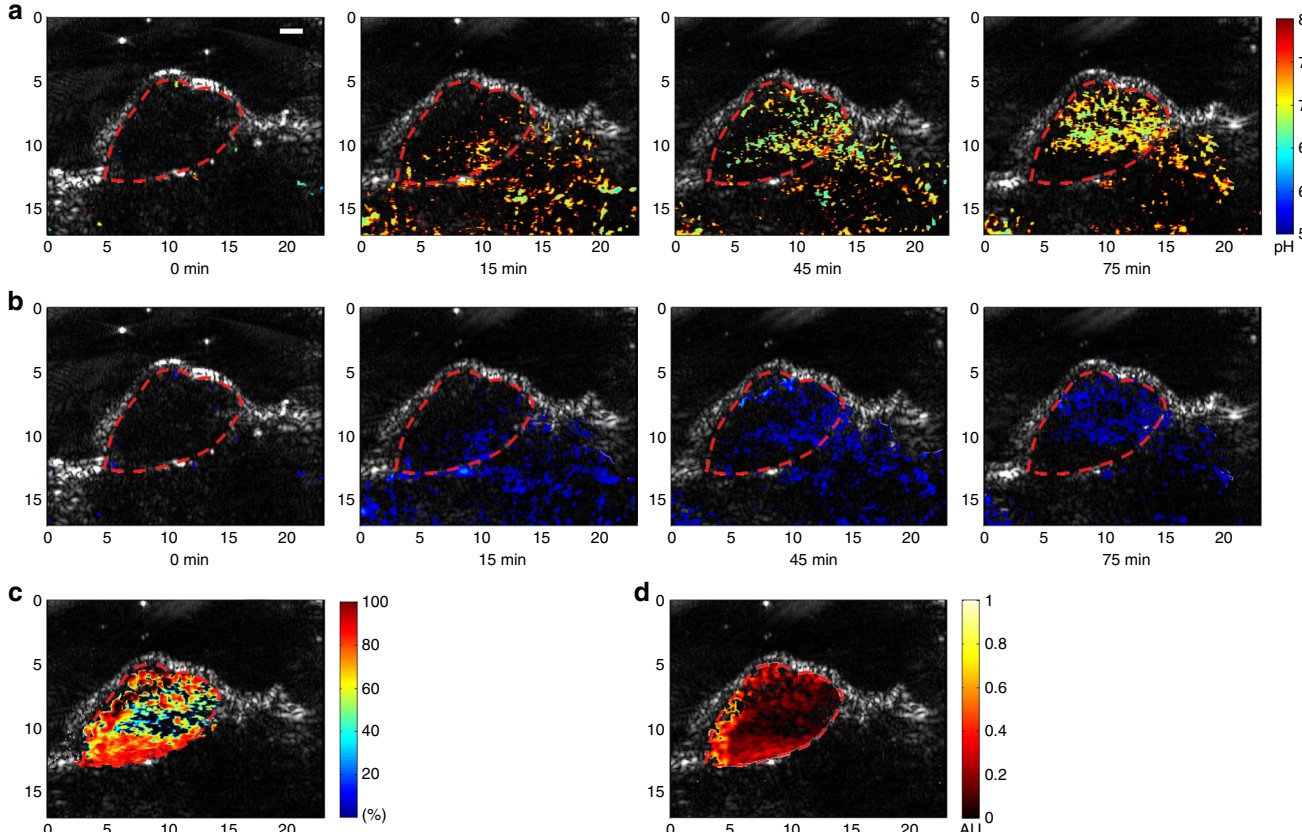

**Fig. 5** Example in vivo quad-wavelength ratiometric PAI of a tumor. Each PA functional image in pseudo-color is superimposed on the gray-scale US image. The tumor area is marked by the dashed line in each image. **a** Quantitative PA pH images at different time points after SNARF-PAA NP injection. *Scale bar*, 2 mm. **b** PA images showing the distributions of SNARF-PAA NPs at different time points after injection. **c** PA image showing the spatially distributed hemoglobin oxygen saturation (sO$_2$) in the tumor area at 75 min after injection. **d** PA image showing the spatially distributed total hemoglobin concentration (THb) in the tumor area at 75 min after injection

i.e., HbO$_2$ and Hb, which have different optical absorption spectra[37]. Therefore, not only the THb but also the hemoglobin oxygen saturation (sO$_2$) affects the spectroscopic PA signal amplitudes from a biological tissue. In simulation, the true pH level was set as 7.4. The errors in pH measurement (i.e., ΔpH) as functions of THb and hemoglobin sO$_2$ were estimated. Since THb affects the ratiometric PA measurement by producing background PA signals, the relative contribution of THb is counted as I$_{THb}$/I$_{NP}$ (i.e., the ratio between the PA signal amplitudes from THb and SNARF-PAA NP). As shown in Fig. 3d, e, not only the presence of hemoglobin but also the hemoglobin sO$_2$ can strongly affect the accuracy in dual-wavelength ratiometric PAI of pH.

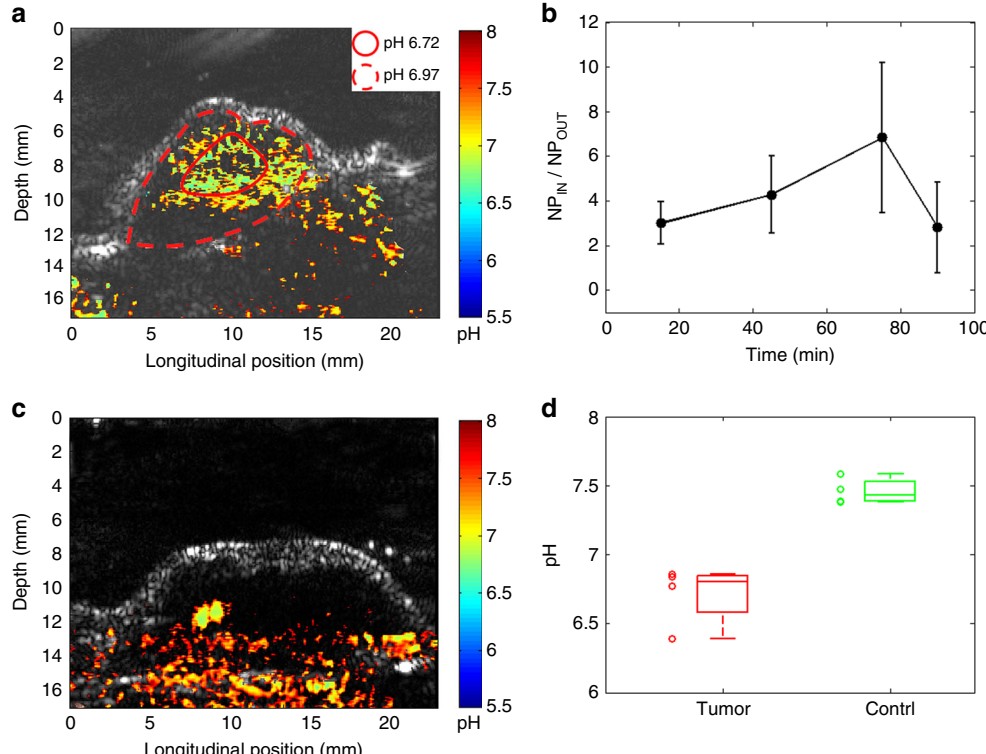

**Fig. 6** Statistical analyses of the PAI results from the in vivo mouse model. **a** A close-up view of the PA pH image of a tumor shown in Fig. 5 (75 min). The pH in the center area (i.e., the area in the solid line) and the peripheral areas (i.e., the area between the solid line and the dash line) are averaged respectively. **b** Analysis of the SNARF-PAA NP accumulation in the tumors at different time points after systemic injection represented by the ratio between the NP concentrations inside and outside the tumor ($NP_{IN}/NP_{OUT}$). With the measurements from four animals ($n = 4$), the average and the standard error for each time point are presented. **c** Example PA pH image of a normal tissue (i.e., thigh), showing relatively higher pH. **d** The boxplot showing the pH levels in tumors ($n = 4$) vs. the pH levels in normal tissues (i.e., thigh) ($n = 4$), as quantified from PA pH images. The measurements from individual sample (tumors and thighs) are also presented by scattered points

The error can be large even when the THb is relatively low compared to SNARF-PAA NP. For example, even when $I_{THb}/I_{NP}$ is lower than 0.1, meaning that the SNARF-PAA NP is still the major optical absorber, the $\Delta pH$ can be larger than 1. This result suggests that, without considering the impact of hemoglobin to spectroscopic PA measurement, quantifying the pH level in biological tissues in vivo is very difficult. For characterizing tumor microenvironment, this is especially challenging, as both the blood content and the blood oxygenation in tumors are largely heterogeneous.

**Quad-wavelength ratiometric PAI of pH**. To overcome the problem faced by dual-wavelength ratiometric PAI, quad-wavelength ratiometric PAI was developed, as described in detail in the Method section. By working with SNARF-PAA NPs buffered at different pH levels from pH 5.8 to 7.8, the PA amplitude ratios between the three wavelengths (576, 584 and 600 nm) and the isosbestic point (565 nm) were measured, as shown in Fig. 4a. The scattered PA measurements for the three wavelengths are linearly fitted, leading to three calibration lines, as marked by $I_{576}/I_{565}$, $I_{584}/I_{565}$ and $I_{600}/I_{565}$, respectively. The three calibration lines enable quantitative measurement of the pH levels by performing quad-wavelength PAI. Figure 4b shows the imaging results from gel phantoms buffered at different pH levels (pH 6.6, 7.0 and 7.4). Although these phantoms were also mixed with whole blood at 1%, the quad-wavelength ratiometric PAI approach successfully quantified the pH levels in each phantom. In addition, no obvious difference can be seen when comparing

the results from the three columns in Fig. 4b, suggesting that the result from the quad-wavelength ratiometric PAI method is independent on the concentration of the SNARF-PAA NPs. This is extremely important for applications in vivo, as the local concentration of SNARF-PAA NPs in the target tissue is difficult to control, and could also be highly heterogeneous.

**In vivo quad-wavelength ratiometric PAI of tumor**. An example of time-dependent functional imaging result of a subcutaneous tumor is shown in Fig. 5. Each PA functional image is super-imposed on the gray-scale US image acquired at the same time using the same acquisition system. Naturally co-registered with the PA image, the US image works well in delineating the tissue structures including the tumor boundary (as marked by the dash line in each image). The PA images in the first line show the spatially distributed SNARF-PAA NPs in the imaged B-scan section at different time points after systemic administration. The NPs were preferentially accumulated in the tumor area gradually, which, we believe, is a combined result of the enhanced perme-ability and retention effect and the tumor-homing F3 peptides. The amount of NP in the tumor reached a peak at ~75 min after injection. Figure 5a shows the spatially distributed pH levels in the imaged section at different time points after injection. As shown in Fig. 5a at 75 min, we can notice that the pH at the center area of the tumor was a little lower than the pH at the peripheral area of the tumor. A close-up view of this pH image at 75 min is shown again in Fig. 6a, where the two red lines (solid and dash) marked the center area and the peripheral area of the

tumor, respectively. The quantitative measurements demonstrate that the averaged pH level in the area within the solid line is $6.72 \pm 0.29$, while the average pH level in the area between the solid line and the dash line is $6.97 \pm 0.35$.

By performing quad-wavelength ratiometric PAI, the spatially distributed hemoglobin oxygen saturation (i.e., blood $sO_2$) and the spatially distributed total hemoglobin concentration (i.e., THb) within the marked tumor area were quantified, as shown in Fig. 5c, d, respectively. The quantified blood $sO_2$ image at 75 min shows that the center of the tumor area had lower blood $sO_2$ compared to the peripheral area, matching our expectation for tumor hypoxia. The THb image at 75 min shows that the center of the tumor also had relatively lower blood content. This example imaging result demonstrates that, by performing quad-wavelength ratiometric PAI, important functional parameters describing the tumor microenvironment, including acidosis and hypoxia, can be quantitatively mapped at the same time.

With the results from four animals, the targeted delivery of SNARF-PAA NPs to the tumors after systemic administration was further evaluated. With pseudo-color PA images showing the distribution of NPs (e.g., the images in Fig. 5b), the average NP concentration in an area can be computed by the number of color pixels divided by the total number of pixels. The accumulation of NPs in a tumor can be evaluated by the ratio between the NP concentrations inside and outside the tumor (i.e., $NP_{IN}/NP_{OUT}$). A larger ratio suggests better targeted delivery. As the result shown in Fig. 6b, the largest ratio appears at 75 min after injection, confirming again that the concentrations of NPs in the tumors reached a peak at this time point. After the imaging experiment, the animal was killed and the accumulations of SNARF-PAA NPs in different organs were studied ex vivo. The results from fluorescence microscopy confirmed again the targeted delivery of NPs to the tumors, as the details described in Supplementary Fig. 7.

An example PA pH image of a thigh is shown in Fig. 6c, which was acquired using the same method for tumor imaging. Compared to the pH images from the tumors, as the example in Fig. 6a, the images from the thighs as normal controls show pH levels that are neutral (around pH 7.4). The average pH levels in the tumor areas and in the normal thigh muscles from four mice were examined ($n = 4$), as shown by the boxplot in Fig. 6d. The average pH in the thigh muscles was $7.46 \pm 0.095$ while the average pH in the tumors was $6.71 \pm 0.22$. A $t$-test was performed with a hypothesis that the pH quantified by the quad-wavelength PA ratiometric imaging cannot differentiate the tumors and the normal thigh muscles. A $P < 0.001$ was reached, suggesting that the pH values in the tumors measured by PAI were significantly different from those in the controls. The imaging findings were confirmed by the measurements from a micro pH electrode (STAR A221, Fisher Scientific, Hampton, NH) as the gold standard. The readings from the electrode placed at multiple positions in the tumors were in the range of pH 6.4–7.0; while the readings from the electrode placed at multiple positions in the thigh muscles were in the range of pH 7.2–7.5.

## Discussion

Although the importance of pH in biological tissue, especially with respect to tumor acidosis, has been noted for many decades, we still lack a practical method for imaging and non-invasively quantifying pH in biological samples in vivo. Here we presented a NP assisted, non-invasive, in vivo PAI technique for quantitative pH imaging of subsurface solid tumors in vivo based on optical spectroscopy. The PA pH nanosensors were fabricated by encapsulating the SNARF-5F dye in the PAA NPs. The NP matrix serves a critical role in chemical imaging in vivo, protecting the

sensing dye molecules from direct interactions with interfering proteins or enzymes as well as enabling specific targeting of tumor cells. For instance, direct interaction with albumin can significantly influence an indicator dye molecule's optical characteristics and reduce its pH sensing capability. The experimental results from the phantoms and the in vivo tumor model demonstrated that quad-wavelength ratiometric PAI allows a differentiation of the optical spectra of hemoglobin and pH-dependent SNARF-PAA NPs. As a result, quantitative imaging of pH without being affected by the background light absorption in biological tissues becomes possible. In addition, the quantification of pH is unaffected by the variation in the SNARF-PAA NP concentration. All these features are crucial for practical applications of the technique in vivo. At the same time, for pH imaging, the quad-wavelength PAI also allows quantitative evaluation of tumor hemodynamic properties, including blood volume and blood oxygenation.

In this study, we assumed that the major chromophores in the target biological samples are the two forms of hemoglobin, which is true in the spectral region of 565–600 nm. In the future, when other chromophores may be present in the target tissue, imaging at additional wavelengths will be needed. However, the number of wavelengths scanned by for PAI is related to the cost of the imaging system as well as the imaging speed. Currently, using a single tunable laser as the light source, the switching between laser wavelengths limits the imaging speed. When a multi-wavelength light source (e.g., a system working with multiple lasers each firing at a different wavelength) or a tunable laser enabling very fast wavelength switching becomes available, two-dimensional (2D) B-scan imaging of tumor pH could be achieved at higher speed or even in real-time fashion. In the future, three-dimensional imaging of the tumor microenvironment is also possible, either by performing a scan of a linear array ultrasound probe or by using a more advanced 2D array ultrasound probe.

The commercially available SNARF-5F is considered as a long-wavelength fluorescent pH indicator, and has been well-developed and widely used for fluorescence based measurement of pH in vitro. This is the reason that the SNARF-5F was employed to build the pH sensing PAA NPs in this proof-of-principle study. Working in the spectral range of 565–600 nm, quantitative imaging of pH by quad-wavelength ratiometric PA measurement can be achieved with satisfactory accuracy in subsurface tissue at a depth up to 6 mm. We have further estimated the error in pH quantification due to the optical attenuation at different wavelengths for quad-wavelength PA ratiometric imaging, as described in Supplementary Fig. 8. As expected, the error becomes larger when the imaging depth increases. However, at 6 mm depth in an optically scattering tissue, the error caused by the optical attenuation was less than 0.16 pH. This error could be further reduced by compensating the wavelength-dependent optical attenuation when simulating the point-by-point pH levels. In another experiment (Supplementary Fig. 9), we have further assessed the sensitivity limit of PA imaging in detecting the SNARF-PAA NPs in subsurface tissue. At the depth of 6 mm in an optically scattering tissue, a SNARF-PAA NP solution at very low concentration of 0.05 mg ml$^{-1}$ can still be detected with a good signal-to-noise ratio (SNR) over 20 dB. The 6 mm depth achieved by PA pH imaging is one order of magnitude deeper than that accessible by fluorescence microscopy. With the current imaging depth, many clinical applications on relatively superficial tumors become possible, such as head and neck cancer, colorectal cancer and cervical cancer, which are all associated with an acidic tumor microenvironment[38–40]. For imaging of deeper tumors, pH indicating dyes that absorb at longer optical wavelengths (e.g., 650–950 nm) will be needed so as

to further improve the optical penetration. One advantage of our PAA NP system is that replacing the dye inside the NP is quite simple.

In oncology, low pH (acidosis) is believed to influence cell metabolism and tumor progression. We believe that our imaging technique has a high potential for cancer research by answering some important general biological or pathological questions. With its many advantages, this non-invasive and non-ionizing imaging technique is also highly translational, and may enable image-guided treatment of cancer in clinical settings. It has been reported that the cytotoxicities of some drugs such as daunorubicil, doxorubicin and mitoxantrone are reduced under acidic condition; while other drugs, such as chlorambucil, cyclophosphamide and 5-fluorouracil, show higher cytotoxicity at lower pH[41]. Therefore, being capable of quantifying the pH level of the tumor microenvironment using imaging technology may enable the optimization of chemotherapy and facilitate personalized treatment for cancer patients. In addition, this nanotechnology based imaging method can readily be applied to other diseases where pH plays an important role. The PA nanosensor described in this work can also be developed for other biologically relevant chemical analytes and thus functional imaging of other physiological parameters (e.g., $K^+$ for hyperkalemia $O_2$ for hypoxia) can be realized using a PAI approach. On top of the sensing capability, other therapeutic agents (i.e., chemotherapy and/or photo-activated therapy) can also be incorporated into such multifunctional NPs[34, 35], turning them into theranostic NPs. Our PA chemical imaging method can thus broaden the field of nano-diagnostics.

## Methods

**Chemicals**. All chemicals were purchased from Sigma Aldrich or ThermoFisher Scientific unless otherwise noted.

**Synthesis of SNARF-PAA NP and F3 surface modification**. SNARF-5F has a pKa ~ 7.2 which is optimal for sensing near neutral physiological pH. The SNARF-5F was encapsulated by polyacrylamide polymers crosslinked by glycerol dimethacrylate (GDMA) to produce SNARF-PAA NPs[29]. All reactions were performed in the dark. Monomer solution containing 1.3 ml of Millipore water, SNARF-5F 5-(and-6)-carboxylic acid (SNARF) (3 mg in 100 µl DMSO), Acrylamide (9.7 mmol), 3-(aminopropyl)methacrylamide hydrochloride salt (APMA) (0.3 mmol) and the crosslinker, GDMA, (2 mmol) was added and emulsified in a surfactant solution of dioctyl sulfosuccinate sodium salt (AOT) (1.6 g) and Brij L4 (3.3 ml) in Hexane (45 ml). The polymerization was initiated by addition of N,N',N'-tetramethylethylenediamine (TEMED) (100 µl) and 10% (w/w) Ammonium Persulfate (100 µl). A small percentage of APMA was introduced as another monomer to provide primary amines used for later surface modification. The reaction was allowed to stir for 2 h and hexane was removed by rotary evaporation. The prepared NPs were washed with ethanol and water with Amicon Filter Cell using a 300 kDa filter, and then lyophilized. Although slight leaching of the dye was observed initially, during purification, there was no leaching during further steps of the synthesis. The surface of the SNARF-PAA NP was PEGylated and conjugated with tumor-homing F3 peptides, following previous protocols[32]. The cancer targeting capability of our PAA NPs conjugated with F3 peptides has been extensively studied[30–32]. Bi-functional polyethylene glycol (MAL-PEG-SCM, 2 kDa, Creative PEGWorks) (4 mg) was added into SNARF-PAA in phosphate-buffered saline (PBS, pH 7.4) (50 mg per 2.5 ml). After 30 min of stirring, it was washed with PBS using Amicon Ultra Centrifugal Filter (100 kDa) and F3 Peptide (KDEPQRR-SARLSAKPAPPKPEPKPKKAPAKKC, RS Synthesis) (11 mg) was added and stirred overnight. Cysteine (0.63 mg) was added and stirred for 2 h to deactivate unreacted maleimide groups. The NP solution was washed with water and lyophilized. Blank PAA NPs were synthesized in the same method without SNARF-5F. The dye loadings of the NPs were approximately 2.4 nmol of SNARF-5F per 1 mg of NPs. The SNARF-PAA NP was characterized by UV–VIS spectroscopy (UV-1601 Spectrometer, Shimadzu), fluorescence spectroscopy (FluoroMax-3, Horiba), and Dynamic Light Scattering instrument (DLS, Delsa Nano C particle analyzer instrument, Beckman Coulter).

**Study of free SNARF-5F and SNARF-PAA NP interactions with albumin using UV-VIS**. Different pH buffers were prepared by mixing different amounts of NaOH and KH₂PO₄. The pH level of each sample was confirmed using the micro pH electrode. To study the free SNARF-5F and the SNARF-PAA NP interactions with proteins, HSA (4 mg ml⁻¹) was introduced into different pH buffers

containing either free SNARF-5F or SNARF-PAA NPs (2 mg ml⁻¹). Free SNARF-5F dye concentration was equivalent to that loaded in SNARF-PAA NPs. The possible change in the optical absorption property of each sample containing either free SNARF-5F or SNARF-PAA NPs caused by interaction with HSA was measured using the UV-1601 Spectrometer (Shimadzu).

**Imaging system**. PAI of phantoms and animals in vivo was performed using our US and PA dual imaging system built on a commercially available research US platform (V1, Verasonics, Redmond, WA) and a linear array probe working at a central frequency of 11.25 MHz (CL15-7, Philips, Andover, MA,). The laser light was from an optical parameters oscillator (Slopo, Continuum, Santa Clara, CA) pumped by an Nd:YAG laser (Surelite, Continuum, Santa Clara, CA) working at a 10 Hz pulse repetition rate and with a pulse width of 5 ns. The details of this imaging system have been introduced in our former publication[24]. Powered by a GPU card, this dual-modality system can acquire PA and US images from the same sample at the same time, both in real-time fashion with a frame rate of 10 Hz (i.e., the laser pulse repetition rate). The fast imaging speed is highly valuable for functional PA imaging, because multi-wavelength images need to be acquired within a relatively short time period before functional parameters potentially change. The quantified spatial resolution for PAI are 226 µm lateral and 166 µm axial at a depth of 6 mm. The light beam on the sample surface formed a rectangle shape with a size of 1.5 cm by 3 cm. The light fluence was less than 20 mJ cm⁻² which, according to the American National Standards Institute (ANSI) standard, is safe for human skin. The good sensitivity of this system in imaging SNARF-PAA NPs in subsurface tissue has been validated, as the details shown in Supplementary Fig. 9. At a very low concentration of 0.05 mg ml⁻¹ (400 times dilution of the injection concentration), an SNR over 20 dB was achieved in imaging SNARF-PAA NPs at 6 mm depth in optically scattering tissue.

**Method of dual-wavelength ratiometric photoacoustic pH measurement**. To obtain the calibration line for dual-wavelength ratiometric PA measurement of pH, SNARF-PAA NP solutions (concentration 20 mg ml⁻¹) with different pH levels from 5.8 to 7.7 (0.1 pH interval) were measured at two wavelengths (i.e., 565 and 600 nm). The pH level of each solution was confirmed by the pH electrode. Illuminated by the laser beam, the PA signals from each solution were collected by a cylindrically focused ultrasound transducer (V312, Panametrics). The signal amplitude at 600 nm divided by the signal amplitude at the isosbestic point of 565 nm gave a ratio. With ratios from three samples at each pH level (n = 3), an average and a standard deviation were obtained.

**Method of quad-wavelength PA ratiometric imaging of pH**. To overcome the problem faced by dual-wavelength ratiometric PAI, a method based on PAI at additional wavelengths, i.e., quad-wavelength ratiometric PAI was proposed. Besides the original two optical wavelengths involved in dual-wavelength ratiometric imaging (i.e., 565 and 600 nm), another two additional wavelengths (i.e., 576 and 584 nm) were introduced to separate the contributions to spectroscopic PA measurement from the two forms of hemoglobin and the SNARF-PAA NP. The wavelength at 584 nm is an isosbestic point where HbO₂ and Hb have the same optical extinction coefficient. At 576 nm, the optical extinction coefficient of HbO₂ has a local peak, enabling high sensitivity in separating the two forms of hemoglobin.

Assuming that the main chromophores in the target sample are HbO₂, Hb and SNARF-PAA NP, then the PA intensity at wavelength $\lambda$ can be expressed as in Equation (1):

$$P_\lambda = k[\varepsilon_{\mathrm{HbO_2}\_\lambda} C_{\mathrm{HbO_2}} + \varepsilon_{\mathrm{Hb}\_\lambda} C_{\mathrm{Hb}} + \varepsilon_{\mathrm{NP}\_\lambda} C_{\mathrm{NP}}], \tag{1}$$

where $k$ is a constant depending on experimental conditions including the Grüneisen parameter of the tissue, the light fluence, and the sensitivity of the imaging system. Assuming that the light fluence, after calibration of the output energy from the laser, are the same for all the optical wavelengths, $k$ will be independent of the wavelength. This assumption holds when the optical wavelengths applied are close and the imaging depth is limited, as discussed in detail in Supplementary Fig. 8. $\varepsilon$ indicates the absorption coefficient of HbO₂, Hb or NP at wavelength $\lambda$, and $C$ is the concentration of HbO₂, Hb or NP. Similar to the optical absorption of SNARF-PAA NPs at 600 nm, the optical absorption of SNARF-PAA NPs at 576 or 584 nm can be expressed as a linear function of the optical absorption at the isosbestic point of 565 nm and the pH value:

$$\varepsilon_{\mathrm{NP}\_\lambda} = \varepsilon_{\mathrm{NP}\_\lambda565}(\alpha \cdot \mathrm{pH} + b) \tag{2}$$

where the constants $\alpha$ and $b$ for each wavelength (576, 584 and 600 nm) can determined by performing ratiometric PA measurement of the SNARF-PAA NPs ex vivo. Therefore, the PA signal amplitudes at all the four wavelengths can be

written as

$$P_{\lambda 565} = k[\varepsilon_{HbO_2\_\lambda 565}C_{HbO_2} + \varepsilon_{Hb\_\lambda 565}C_{Hb} + 0.\text{pH}\varepsilon_{NP\_\lambda 565}C_{NP} + 1 \cdot \varepsilon_{NP\_\lambda 565}C_{NP}]$$

(3)

$$P_{\lambda 576} = k\left[\varepsilon_{HbO_2\_\lambda 576}C_{HbO_2} + \varepsilon_{Hb\_\lambda 576}C_{Hb} + \frac{a_{\lambda 576}}{I_{\lambda 565}}\text{pH}\,\varepsilon_{NP\_\lambda 565}C_{NP} + \frac{b_{\lambda 576}}{I_{\lambda 565}}\varepsilon_{NP\_\lambda 565}C_{NP}\right]$$

(4)

$$P_{\lambda 584} = k\left[\varepsilon_{HbO_2\_\lambda 584}C_{HbO_2} + \varepsilon_{Hb\_\lambda 584}C_{Hb} + \frac{a_{\lambda 584}}{I_{\lambda 565}}\text{pH}\,\varepsilon_{NP\_\lambda 565}C_{NP} + \frac{b_{\lambda 584}}{I_{\lambda 565}}\varepsilon_{NP\_\lambda 565}C_{NP}\right]$$

(5)

$$P_{\lambda 600} = k\left[\varepsilon_{HbO_2\_\lambda 600}C_{HbO_2} + \varepsilon_{Hb\_\lambda 600}C_{Hb} + \frac{a_{\lambda 600}}{I_{\lambda 565}}\text{pH}\,\varepsilon_{NP\_\lambda 565}C_{NP} + \frac{b_{\lambda 600}}{I_{\lambda 565}}\varepsilon_{NP\_\lambda 565}C_{NP}\right].$$

(6)

Equations (3)–(6) can be converted to matrix form

$$k \cdot \begin{bmatrix} \varepsilon_{HbO_2\_\lambda 565} & \varepsilon_{Hb\_\lambda 565} & 0 & \varepsilon_{NP\_\lambda 565} \\ \varepsilon_{HbO_2\_\lambda 576} & \varepsilon_{Hb\_\lambda 576} & \frac{a_{\lambda 576}}{I_{\lambda 565}}\varepsilon_{NP\_\lambda 565} & \frac{b_{\lambda 576}}{I_{\lambda 565}}\varepsilon_{NP\_\lambda 565} \\ \varepsilon_{HbO_2\_\lambda 584} & \varepsilon_{Hb\_\lambda 584} & \frac{a_{\lambda 584}}{I_{\lambda 565}}\varepsilon_{NP\_\lambda 565} & \frac{b_{\lambda 584}}{I_{\lambda 565}}\varepsilon_{NP\_\lambda 565} \\ \varepsilon_{HbO_2\_\lambda 600} & \varepsilon_{Hb\_\lambda 600} & \frac{a_{\lambda 600}}{I_{\lambda 565}}\varepsilon_{NP\_\lambda 565} & \frac{b_{\lambda 600}}{I_{\lambda 565}}\varepsilon_{NP\_\lambda 565} \end{bmatrix} \begin{bmatrix} C_{HbO_2} \\ C_{Hb} \\ \text{pH} \cdot C_{NP} \\ C_{NP} \end{bmatrix} = \begin{bmatrix} P_{\lambda 565} \\ P_{\lambda 576} \\ P_{\lambda 584} \\ P_{\lambda 600} \end{bmatrix}$$

(7)

$$k \cdot \begin{bmatrix} C_{HbO_2} \\ C_{Hb} \\ \text{pH} \cdot C_{NP} \cdot \varepsilon_{NP\_\lambda 565} \\ C_{NP} \cdot \varepsilon_{NP\_\lambda 565} \end{bmatrix} = \begin{bmatrix} \varepsilon_{HbO_2\_\lambda 565} & \varepsilon_{Hb\_\lambda 565} & 0 & 1 \\ \varepsilon_{HbO_2\_\lambda 576} & \varepsilon_{Hb\_\lambda 576} & \frac{a_{\lambda 576}}{I_{\lambda 565}} & \frac{b_{\lambda 576}}{I_{\lambda 565}} \\ \varepsilon_{HbO_2\_\lambda 584} & \varepsilon_{Hb\_\lambda 584} & \frac{a_{\lambda 584}}{I_{\lambda 565}} & \frac{b_{\lambda 584}}{I_{\lambda 565}} \\ \varepsilon_{HbO_2\_\lambda 600} & \varepsilon_{Hb\_\lambda 600} & \frac{a_{\lambda 600}}{I_{\lambda 565}} & \frac{b_{\lambda 600}}{I_{\lambda 565}} \end{bmatrix}^{-1} \begin{bmatrix} P_{\lambda 565} \\ P_{\lambda 576} \\ P_{\lambda 584} \\ P_{\lambda 600} \end{bmatrix}.$$

(8)

The left side of Equation (8) as a $4 \times 1$ matrix can be computed when all the variables on the right side of the equation are known or can be determined by multi-wavelength PA measurements. Then blood $sO_2$ and the pH level of each pixel in the imaging plane can be computed by

$$sO_2 = \frac{kC_{HbO_2}}{k(C_{HbO_2} + C_{Hb})}$$

(9)

$$\text{pH} = \frac{k \cdot \text{pH} \cdot C_{NP} \cdot \varepsilon_{NP\_\lambda 565}}{k \cdot C_{NP} \cdot \varepsilon_{NP\_\lambda 565}}$$

(10)

The images showing the relative distribution of THb and the relative distribution of the SNARF-PAA NP can be computed by

$$\text{THb} = k \cdot (C_{HbO_2} + C_{Hb})$$

(11)

$$\text{NP} = C_{NP} \cdot \varepsilon_{NP\_\lambda 565}$$

(12)

where both $k$ and $\varepsilon_{NP\_\lambda 565}$ are constants. MATLAB (R2010a, Mathworks, Natick, MA) was used for simulation of the results from the phantoms and the animals.

Before quad-wavelength ratiometric PAI can be performed, we first need to obtain the calibration lines for 576 and 584 nm wavelengths, respectively, in addition to the calibration line for 600 nm as shown in Fig. 3b. Following the same procedure, SNARF-PAA NP (concentration 20 mg ml$^{-1}$) with pH from 5.8 to 7.7 (0.1 pH interval) was measured at the wavelengths of 565, 576 and 584 nm. The PA signal amplitudes at 576 and 584 nm were divided by the signal amplitude at the isobestic point of 565 nm. With measurements from three samples at each pH level ($n=3$), averages and standard deviations were obtained. The calibration lines here (Fig. 4a) were obtained using the well-calibrated single-element V312 transducer, benefiting from its excellent receiving sensitivity. However, the calibration lines are independent of the detection system, and also work for the PAI system based on the Verasonics US platform (Supplementary Fig. 10).

We intentionally selected the four optical wavelengths that are close so that the optical spectral range for quad-wavelength PA ratiometric imaging is relatively small (565–600 nm). In this case, when the incident light energy on the sample surface can be calibrated for each wavelength, the distributions of the light fluence in the tissue can be considered similar for all the wavelengths. Otherwise, largely separated wavelengths can lead to significant difference in optical attenuation in tissue, which, if not compensated, can affect the accuracy in quantifying tumor pH using quad-wavelength PA ratiometric imaging. In other words, the optical spectrum selected needs to differentiate the optical spectra of HbO$_2$, Hb and pH-dependent SNARF-PAA NPs; while the optical attenuation in tissue cannot be largely different within the selected spectrum. We have further studied the potential error in pH quantification

due to the spectroscopic difference in optical attenuation in tissue, as the details shown in Supplementary Fig. 7. At a depth of 6 mm in optically scattering tissue, the potential error caused by the difference in optical attenuation over the spectral range of 565–600 nm is ~0.16 pH. This already minor error can be further reduced by compensating the light attenuation in the simulation.

**PAI on phantoms**. For the studies on phantoms, including both dual-wavelength and quad-wavelength ratiometric imaging, the phantoms were made from porcine gel (concentration 80 g l$^{-1}$). Each phantom contained SNARF-PAA NPs with a concentration of 20 mg ml$^{-1}$, and was buffered at different pH levels (pH 6.6, 7.0 and 7.4). For dual-wavelength ratiometric imaging, the PA image of a phantom at 600 nm, after being smoothed by Gaussian filter, was divided by the PA image at 565 nm, which led to a pixel-by-pixel PA ratiometric image of the phantom. For quad-wavelength ratiometric imaging, each phantom was imaged with the four wavelengths. Then the 2D pH image was computed following the procedure described. To study the potential error in pH imaging caused by the background optical absorption, whole blood was added in some phantoms with a concentration of 1% (w/w). 1% blood content is reasonable, considering that, as reported in a literature, the blood content in human glioma tumor ranges from 0.95 to 2.79%[42]. To find out whether the pH measurement based on quad-wavelength ratiometric imaging is affected by the concentration of the SNARF-PAA NPs, phantoms were also made with different NP concentrations (2, 10 and 20 mg ml$^{-1}$).

**PAI of tumor pH in a mouse model in vivo**. All the procedures on live animals were approved by the University Committee on the Use and Care of Animals (UCUCA) of the University of Michigan (U-M). The mice were housed at the U-M Medical School in the Unit for Laboratory Animal Medicine (ULAM). In total, four mice (5 weeks old male, Athymic nude Fox/NU, Envigo) were used. 9 l rat glioma cell line (American Type Culture Collection) was cultured in RPMI 1640 medium supplemented with 10% fetal bovine serum (FBS) and 1% antibiotic-antimycotic. Approximately 10$^6$ cells in 100 μl of culture media were subcutaneously injected on the back of each mouse. Each tumor was allowed to grow for 2–3 weeks until its volume reached ~0.5 cc as measured using a caliper.

The nude mice were anesthetized with the inhalation of 1.0–2.0% isoflurane mixing with oxygen. During imaging, the laser light covered the entire tumor area as well as some surrounding tissue. The ultrasound probe was fixed around 1 cm above the tumor. During imaging experiment, the body temperature of mouse was also maintained with a heating lamp. The SNARF-PAA NP solution (20 mg ml$^{-1}$ in saline) was injected through tail-vein (250 mg NP per kg body weight). The PA images were acquired before the injection (0 min), and at 15, 45 and 75 min, respectively after the injection. To improve the SNR, the PA images acquired at each wavelength were averaged 50 times. Considering that the wavelength switching time was 5–10 s, the total time period for PA imaging at the four wavelengths was less than 60 s. To avoid potential motion artifact, the animal especially the tumor area under the scan was fixed tightly during image acquisition using a home-fabricated fixation device. With the images from the four wavelengths acquired, the spatially distributed NP concentrations and the spatially distributed pH levels were computed for each time point. For image pixels with weak PA signals, calculating the ratios among different wavelengths, as performed in quad-wavelength PA ratiometric imaging, could have large errors caused by the background noise. Therefore, to reduce the possible errors, a universal threshold of 6 dB above the background noise level was set for PA intensity images, meaning that the pixel-by-pixel computation of the pH level was performed only for those pixels with intensities above the 6 dB threshold.

To provide a control, quad-wavelength PAI was also performed on normal tissues. For each mouse, the SNARF-PAA NP solution (20 mg ml$^{-1}$ in saline) was subcutaneously injected into the thigh (50 mg NP per kg body weight). Since the distribution of the locally injected NPs in the thigh is relying on perfusion, instead of a single injection of a high dose, multiple spatially scattered injections each with smaller dose were applied aiming at achieving more homogenous distribution of NPs in the thigh. The PA images of the treated thigh were obtained at 10 min after the injection. The PA images at the four wavelengths were used to compute the spatially distributed pH levels in the thigh. After the imaging experiment was finished, the pH levels in the tumors and the thighs were measured by inserting the micro pH electrode in multiple positions in the target tissues.

**Data availability**. The data that support the finding of this study are available from the corresponding authors on reasonable request.

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

## Acknowledgements

We acknowledge Kimberly Ives for assistance in writing and revising of the Animal Protocol, Cheng Xu for helpful discussions on animal handling, the staff of the Microscopy & Image Analysis Laboratory at the University of Michigan for their assistance, Jeff Harrison for helping with the Transmission Electron Microscope, and Bruce Donohoe for helping with the Fluorescence Microscope. This work was supported by NIH/NCI under the grant number R01CA186769 (R.K. and X.W.).

## Author contributions

R.K. and X.W. are corresponding authors. R.K. and X.W. conceived and supervised the project. J.J., C.H.L., R.K. and X.W. designed the experiments. C.H.L. prepared the nanoparticles. J.J. and C.H.L. performed the in vitro and in vivo photoacoustic experiments. J.J. performed the data analysis and imaging processing. J.J., C.H.L., R.K. and X.W. wrote the manuscript.

## Additional information

**Competing interests:** The authors declare no competing financial interests.

