## [Peer Review file · Nature Communications]

Reviewers' comments:

Reviewer #1 (Remarks to the Author):

The manuscript by Wang et al. has described a novel study using multi-spectroscopic photoacoustic imaging (PAI) to quantify the pH microenvironment in vivo. This is a very timely work that has demonstrated the marriage between a cutting-edge imaging technology and a well-matched biomedical problem, enhanced by advancement in materials science. It is a multi-disciplinary research that can maximize the translational potential of this promising imaging technology.

From the technological perspective, it has been a long-standing problem that the existing imaging technologies cannot provide noninvasive measurement of pH in vivo. The electrode-based pH-meter is typically invasive and can only measure a limited number of points. PAI is a highly promising technology that can penetrate deep into tissue with optical absorption contrast.

However, the previous attempt of using PAI for pH measurement has failed for in vivo settings, because the background signals from blood always overshadow the signals from the pH-sensitive dyes. The major contributions of this work include (1) a nanoparticle-based pH sensing contrast agent that is not subjected to the endogenous proteins in vivo, and (2) a ratio-based PAI method that can robustly separate the weak signals from the pH dye and the signals from blood. Both contributions are significant steps towards a new platform for in vivo pH sensing. Moreover, this manuscript has also opened a new window for nanoparticle-PAI-based measurement of other biological parameters beyond pH, which can certainly adapt a similar strategy.

With that said, in order to publish at NC, I do believe that the authors have to strengthen the manuscript by adding more technological details and better justify several key questions, as listed below.

Major concerns:

1. It is a smart idea that the authors encapsulated SNARF-5F into nanoparticles. However, it was not reported in the manuscript whether the nanoparticle itself has any significant absorption at the same absorbing wavelengths of SNARF-5F. If yes, the authors need to consider correcting for the nanoparticle's absorption, which may contribute to the error of the final pH measurement. Moreover, what is the photostability of the nanoparticle and the dye (photobleaching or not)?
2. As the authors have correctly studied, one concern of encapsulating the SNARF-5F into nanoparticles is the fidelity of free exchange of ions with the outside environment. In addition to showing that the optical spectra indeed differ with the pH, I think the authors need to do a more detailed study about the kinetics of the pH sensing. For example, how long does it take for the encapsulated dyes to fully access the ions in the environment? Does the encapsulation reduce the ion diffusion efficiency? Does the high-concentration of SNARF-5F inside the nanoparticle induce any change to the optical absorption (like ICG)?
3. The authors have thoroughly investigated the improvement of four-wavelength measurement over two-wavelength measurement. However, the authors may want to discuss the potential errors induced by the wavelength-dependent optical attenuation at the four wavelengths, since the wavelength difference is not negligible? Optical fluence should be included in Eq. (1-4). Ratiometric measurement will not compensate for the optical fluence.

Minor concerns:

1. Line 65. It is not clear if the authors are talking about fluorescence lifetime based pH measurement? Also, if autofluorescence is an issue for fluorescence-based method, it will likely be an issue for PAI as well, since they are both based on optical absorption.
2. A justification of the selected wavelengths will be helpful. Why is 600 nm chosen here? Is it the optimal wavelength providing the best sensitivity to the pH difference? Same to other two wavelengths used in the work.
3. The authors may also want to add a simple explanation about the underline mechanism of pH sensing using SNARF-5F.
4. Line 142, the authors may want to justify the linear fitting in the ratiometric measurement. It seems to me that a higher order fitting is more accurate. The same question is for Fig. 4a.
5. Please quantify the measurement errors (or accuracy) in Fig. 4b.

6. Line 194, what is the standard deviation of the measurements in vivo? Are the difference between the two measurements larger than the measurement accuracy quantified in Fig. 4b?
7. Fig. 5, how is the ROI of the tumor region selected?
8. Fig. 5, please discuss if the measurement accuracy would decrease with the penetration depth?
9. Line 297, if a pixel-by-pixel ratio is calculated, will some pixels with weak signals give large errors due to noise? i.e., it is always a dangerous operation to divide two small numbers? Did the authors performed any thresholding before calculating the ratio?
10. Fig. 6c, why is the particle distribution so imhomogenous if there is no tumor?
11. The authors may consider citing the previous work using SNARF-5F as PAI contrast for pH sensing, by Chatni et al.. Journal of Biomedical Optics 16, 100503
12. It would be helpful if the authors can add more details of their PAI system, such as ultrasound frequency, optical fluence, and image reconstruction method.
13. Fig. 6d, is the pH value in the tumor significantly different from that in the control?

Reviewer #2 (Remarks to the Author):

In the paper entitled "In vivo quantitative imaging of tumor pH: Nanosonophore assisted multi spectral photoacoustic imaging", the authors introduce a technique to quantitatively measure tumor pH in vivo using tumor-targeted pH sensing nano-probes and an imaging modality that combines photoacoustics and ultrasound. The authors used pH sensitive nanosonophores, which are encapsulated optical pH indicator (SNARF-5F) in polyacrylamide nanoparticles, and added surface modification for both tumor targeting and avoidance of the immune system. The authors used different wavelengths in PA imaging to separate the contrast agents signal from oxy and deoxy hemoglobin, and thereby calculate the pH in tumor environment. The method was demonstrated both in vitro and in vivo.

Overall, the results in the paper are adequately presented. However, I think that for Nature Communication, the novelty of the paper is very marginal and not sufficient. In particular, compared to the group's previous published work in this field (Ray et al., 2013, Analyst, "Sonophoric nanoprobe aided pH measurement in vivo using photoacoustic spectroscopy), which reports both in vitro and in vivo pH measurements, including "successful pH sensing in vivo on a rat joint model, with a precision better than 0.1 pH units" and the work on tumor targeted nanoparticles (Ray et al., 2011, Nano Res., "Targeted blue nanoparticles as photoacoustic contrast agent for brain tumor delineation"). The modification the group performed to their previous work is the use of four wavelengths rather than two to extract the concentration of the pH sensitive dye more accurately, and thereby quantify the pH. Extraction of dyes in whole blood using multiple wavelengths has already been reported (e.g., Li et al., 2008, Proceedings of IEEE 96, "Simultaneous molecular and hypoxia imaging of brain tumors in vivo using spectroscopic photoacoustic tomography").

Therefore, I think that the work is not suitable for publication in Nature Communication. In addition, I have the following comments and questions:

1. The authors suggest that the imaging technology reported may be used to monitor "cancer microenvironment in vivo, WITH IMPORTANT RAMIFICATION FOR THE OPTIMAL CHOICE OF CHEMOTHERAPY" – However, the results don't support such a claim.
2. In figure 3, the authors show how the total hemoglobin concentration as well as the oxygen saturation affects the ratio-metric method for calculating the pH. I think that these results are obvious and expected considering that when one would like to extract three parameters (HbO₂, Hb, and pH) at least three independent equations are required, and therefore the use of two wavelengths is not sufficient. In other words, the authors should explain why the dual-wavelength ratiometric method was originally suggested to work in whole blood environment, in which the oxygen saturation is unknown.
3. The phantom experiments of the quad-wavelength ratiometric method were done with 1% of whole blood (Line 177). It seems to me that 1% of whole blood diluted in 99% of buffer is not

sufficient to prove that the method can efficiently separate the pH from whole blood in vivo. Moreover, the main assumption in using the linear equations to extract the concentrations of different absorbers is linearity. It was previously shown that spectroscopic PA measurements are affected by nonlinearity of the blood absorption (e.g., Danielli et al., 2015, APL, "Nonlinear photoacoustic spectroscopy of hemoglobin"). Have the authors verified that in whole blood the PA signal is linear with incident power and that the extraction of the dye is accurate?

4. In Methods, The PA system is not detailed. What is the laser fluence at the surface, what is the resolution and imaging depth of the system?

Reviewer #3 (Remarks to the Author):

The manuscript describes a method of measuring pH in the tumor non-invasively. Although the approach of using photoacoustics for pH measurements is not quite unique, the results are promising. Overall, the manuscript requires some additional input from the authors as specified below.

1. The authors used a long-standing and commercially available ratiometric pH indicator SNARF. One of the reasons that this dye has not been used in vivo is due to strong attenuation by both scattering and tissue absorption. The general approach of using red dyes for imaging does not go beyond the small animal models with subcutaneous tumors. Even this model requires relatively large amount of the probe as the authors used (250 mg NP, which is an equivalent of 17 g of human injection). Please comment. Can the authors provide an estimate of how far the signal can penetrate at 560-600 nm?

2. A large dosage might cause a rapid change in the metabolism that might lead to the change in the pH of the tissue. Please comment.

3. How do authors see the application of the proposed method for the pH measurements? According to the provided results the pH inside the tumor is 6.7 which is within the range of normal pH level in the body. In that case and using the proposed method the location of the tumor might never be found. Please describe how the probes can be used for diagnostics or drug selection as mentioned in the discussion?

4. Did the authors perform any blocking studies to verify the uptake by the tumor? Does targeting with F3 work or that is another example of the EPR effect?

5. Line 57 – 60. The manuscript only mentions MRI and PET. However, there is a large effort of developing pH optical probes primarily with NIR dyes that is not mentioned in the manuscript: Berezin et al Biophys. J, 2011, 20; 100(8):2063-72., Lee et al Photochem Photobiology, 2013, 89 (2) 326–331, Gilson et al Mol. Pharmaceutics, 2015, 12 (12), pp 4237–4246, Chen et al Advanced materials , 2015 6820-6827 and some others. Please provide mor background on using optical probes for in vivo

6. Line 120-125. Often the dyes change their absorption and fluorescent properties in the presence of albumin that causes the shift of the isosbestic point due to a hydrophobic environment. Therefore ratiometric approach might still work but for a different set of wavelengths. Provide the abs spectra of the dye in the presence of albumin at different pH and comment.

Reviewer #4 (Remarks to the Author):

Review: Jo, Lee, Kopelman, Wang. "In vivo quantitative imaging . . ."

Overall description and comments:

The authors describe problems with existing approaches for quantitative *in vivo imaging* of pH and its importance for cancer diagnostics and monitoring a response to therapy. Their proposed solution is a smart nanoplatform that contains a standard pH-sensitive dye (SNARF 5F), which can be functionalized to target cancerous tissue. Their rationale is that the encapsulated dye is more stable than the free dye, whose response changes in the presence of tissue protein/albumin. Instead of fluorescent imaging of the dye, however, the authors employ photoacoustic imaging with spectral unmixing to map total hemoglobin and tissue pH at depth and with good resolution (<200 μm). Through careful and methodical testing in photoacoustic phantoms, they develop calibration curves for sensing pH that are independent of the concentration of the dye or effects of hemoglobin/sO₂, which is often a confounding factor during imaging of tumors. Finally, they demonstrate proof-of-concept of *in vivo* pH imaging in tumors on the back of 4 mice using a 9L rat glioma cell line. The thigh was used as a control. They also describe clearance of the agents and give support that the nanoplatform is biocompatible. The study is quite novel and offers the cancer research community a new type of contrast agents that others can employ to map pH noninvasively for their small animal research. As long as the investigator has access to photoacoustic imaging and spectroscopy at visible wavelengths, they should be able to implement the agent and algorithms into their research. Although the penetration at visible wavelengths of a few millimeters is limited due to high scattering and absorption, this might be sufficient for orthotropic tumor imaging in mice. As a new noninvasive tool for imaging and sensing, the techniques could play an important role in developing new diagnostic techniques and therapies for cancer. The manuscript is well written with rationale arguments, provides sufficient background, and details the methodology in comparison with standard techniques. The methods section clearly describes the unmixing algorithm to separate background absorption (Due to hemoglobin) from the pH signal. Because this is a proof-of-concept study, it is acceptable that the authors have a relatively low # of animal subjects (4 mice) and trials for averaging across experiments to determine the calibrations (in some cases as few as an n=3). This was fine as the initial study in phantoms and mice. However, there are still some areas that need improvement, which are described below.

Major Comments:

1) There is very little discussion on the limitations of using the nanoagents for photoacoustic imaging and pH sensing at visible wavelengths. What is the penetration limit due to scattering and absorption? How does the accuracy change with depth in a highly scattering/absorbing environment? What about effects from background absorbers in tissue other than hemoglobin? What was the degree of photobleaching during the experiments (e.g., loss of PA signal per pulse normalized to fluence)?

2) It appears that the phantom work was done at room temperature. Did the investigators look at the effects of temperature (e.g., body temperature) and whether that affects the calibration curve in a significant or measureable way?

- 3) Fig 2a employed a different setup (e.g., V312) than the Fig 2b and the other experiments (Verasonics). Was there a reason that the Verasonics system could not be used for all of the experiments for a fair comparison?
- 4) Was a breathing artifact apparent during mouse imaging? How was this handled as a source of error during spectroscopy/unmixing.
- 5) Can the authors add a bit more detail on how the light was delivered to the mouse? Was it through a dual fiber bundle? What was the energy out of the laser and fluence on the skin? How much averaging was performed (reps per wavelength)? What was the wavelength switching time?
- 6) Was 3D imaging performed (or possible) with your setup?
- 7) Are there any potential human applications of the technology given the limited penetration of a few millimeters? What are the current challenges/limitations that need to be overcome?

Minor Comments:

- 1) Figure caption/legend for Figure 3 (page after references): Title does not seem to relate to the plots/subplots in this figure. There are no “mechanisms” described in Figure 3. Please restate.
- 2) Page 1 (intro). It is stated that MR and CT require contrast agents for pH mapping. CEST MRI does not necessarily require contrast agents, since it measures changes in water saturation.
- 3) Paragraph from Lines 70-83 is poorly written with many grammatical errors. A few are listed below
“none of these studies were able to ...” → “none of these studies was able to . . .”
Line 72: “. . . resolution is no longer limited by the detection . . .”
Lines 72-74: sentence is poorly phrased; restate: “. . . photons, which suffers from high scattering in soft tissue, . . .”
“Thus, the spatial resolution . . .”
- 4) A few times (e.g., see Fig. 2 caption) “HSA” is misspelled “HAS” Please do a “search and replace.”

Responses to Reviewers' Comments

We greatly appreciate the very valuable and constructive comments from all the reviewers which give us the opportunities to improve this manuscript. This manuscript has been revised accordingly. Additional details and many new results have been included. Below are our answers to reviewers' questions.

REVIEWER #1:

The manuscript by J et al. has described a novel study using multi-spectroscopic photoacoustic imaging (PAI) to quantify the pH microenvironment in vivo. This is a very timely work that has demonstrated the marriage between a cutting-edge imaging technology and a well-matched biomedical problem, enhanced by advancement in materials science. It is a multi-disciplinary research that can maximize the translational potential of this promising imaging technology.

From the technological perspective, it has been a long-standing problem that the existing imaging technologies cannot provide noninvasive measurement of pH in vivo. The electrode-based pH-meter is typically invasive and can only measure a limited number of points. PAI is a highly promising technology that can penetrate deep into tissue with optical absorption contrast. However, the previous attempt of using PAI for pH measurement has failed for in vivo settings, because the background signals from blood always overshadow the signals from the pH-sensitive dyes. The major contributions of this work include (1) a nanoparticle-based pH sensing contrast agent that is not subjected to the endogenous proteins in vivo, and (2) a ratio-based PAI method that can robustly separate the weak signals from the pH dye and the signals from blood. Both contributions are significant steps towards a new platform for in vivo pH sensing. Moreover, this manuscript has also opened a new window for nanoparticle-PAI-based measurement of other biological parameters beyond pH, which can certainly adapt a similar strategy.

With that said, in order to publish at NC, I do believe that the authors have to strengthen the manuscript by adding more technological details and better justify several key questions, as listed below.

Major concerns:

Questions 1. *It is a smart idea that the authors encapsulated SNARF-5F into nanoparticles. However, it was not reported in the manuscript whether the nanoparticle itself has any significant absorption at the same absorbing wavelengths of SNARF-5F. If yes, the authors need to consider correcting for the nanoparticle's absorption, which may contribute to the error of the final pH measurement. Moreover, what is the photostability of the nanoparticle and the dye (photobleaching or not)?*

Answer: The blank PAA NP solutions are optically clear, as shown in the newly added **Fig. S2** in Supplementary Materials. As checked by the UV-VIS spectrophotometer, the optical spectrum of blank PAA NP solution is significantly weaker than that of SNARF-PAA NPs in the wavelength range of 565-600 nm, as shown in Fig. S2b. This weak extinction spectrum of blank PAA NPs measured by the UV-VIS, instead of reflecting their optical absorption, should mostly come from their optical scattering of the lower wavelength light. More importantly, since we took the calibration of the SNARF-PAA NPs (with the potential background absorption, or rather scattering, from NP matrix), possible absorption of blank PAA NPs, if any, will not influence later quantitative imaging of pH levels.

An experiment was performed to study the photostability of SNARF-PAA NPs, as shown in the newly added **Fig. S5** in Supplementary Materials. The SNARF-PAA NP solution was continuously illuminated with a laser beam over a total time period of 60 minutes, and the optical absorption of the solution was measured at different time points. At 15 min after illumination, the change in optical absorption is about 2%. This slow speed of photobleaching is highly desirable for potential in vivo applications. Using our current system, quad-wavelength ratiometric PAI of a tumor takes less than 1 minute for image acquisition at all the four wavelengths. The estimated photobleaching during this time period is less than 0.2%. In addition, the ratiometric method significantly reduces potential errors from photobleaching

because, as we have demonstrated in the manuscript, the pH measurement from the quad-wavelength ratiometric imaging is independent of the NP concentration (**Fig. 4b**). Therefore, we don't expect that photobleaching, which has been proved to be very slow, affects the accuracy of our pH imaging method.

Questions 2. *As the authors have correctly studied, one concern of encapsulating the SNARF-5F into nanoparticles is the fidelity of free exchange of ions with the outside environment. In addition to showing that the optical spectra indeed differ with the pH, I think the authors need to do a more detailed study about the kinetics of the pH sensing. For example, how long does it take for the encapsulated dyes to fully access the ions in the environment? Does the encapsulation reduce the ion diffusion efficiency? Does the high-concentration of SNARF-5F inside the nanoparticle induce any change to the optical absorption (like ICG)?*

Answer: In response to this question regarding sensing kinetics, we have conducted a time-dependent fluorescence measurement, as shown in the newly added **Fig. S4** in the Supplementary Materials. The SNARF-PAA NPs show a quick response (within 1 second) to the change of environmental pH. This result, besides validating the quick kinetics of the SNARF-PAA NPs, also confirmed that our pH nanosensors are fully reversible in sensing the pH change.

The Kopelman group has a long history (~20 yrs) of preparing nanosensors (called PEBBLE) for biological studies. The sensor response time can vary depending on the nanoparticle composition and matrix; however, all the optical sensors respond to local ion changes within a few seconds (R. Kopelman et al., "Optochemical nanosensor PEBBLEs: photonic explorers for bioanalysis with biologically localized embedding", *Curr Opin Chem Biol*, 2004, 5, 540.). Based on our experience as well as the result in **Fig. S4**, we believe that the nanoparticle matrix does not influence much the ion diffusion efficiency.

Indeed, the high-concentration of SNARF-5F inside the nanoparticle induces change to the optical absorption. The newly added **Fig. S1a** in Supplementary Materials shows the absorption spectra of free SNARF-5F at different pH levels; while **Fig. 3a** shows the optical spectra of the SNARF-PAA NPs. They are different. This is a well-known phenomenon when encapsulating dyes into NPs (similar to ICG).

Questions 3. *The authors have thoroughly investigated the improvement of four-wavelength measurement over two-wavelength measurement. However, the authors may want to discuss the potential errors induced by the wavelength-dependent optical attenuation at the four wavelengths, since the wavelength difference is not negligible? Optical fluence should be included in Eq. (1-4). Ratiometric measurement will not compensate for the optical fluence.*

Answer: We intentionally selected the four optical wavelengths to be close, so that the optical spectral range for quad-wavelength PA ratiometric imaging is relatively small (565-600 nm). In this case, when the incident light energy on the sample surface can be calibrated for each wavelength, the distributions of the light fluence in the tissue can be considered similar for all the wavelengths. Otherwise, largely separated wavelengths can lead to significant differences in optical attenuation in tissue, which, if not compensated, can affect the accuracy in quantifying tumor pH using quad-wavelength PA ratiometric imaging. In other words, the optical spectrum selected needs to differentiate the optical spectra of HbO₂, Hb and pH-dependent SNARF-PAA NPs; while the optical attenuation in tissue cannot be largely different within the selected spectrum. This discussion has been added in the Materials and Methods section (**Page 16-17**).

In addition, we have further studied the potential error in pH quantification due to the spectroscopic optical attenuation in tissues, as shown in the newly added **Fig. S8** in Supplementary Materials. At 6 mm depth, which was the maximum tumor depth for the animal model involved in this study, the estimated error caused by the difference in optical attenuation over the spectral range of 565-600 nm is about 0.16

pH. This already minor error can be further reduced by compensating for the light attenuation in the simulation, which, however, was not implemented in this proof-of-concept study.

Minor concerns:

Question 1. Line 65. It is not clear if the authors are talking about fluorescence lifetime based pH measurement? Also, if autofluorescence is an issue for fluorescence-based method, it will likely be an issue for PAI as well, since they are both based on optical absorption.

Answer: We were not talking about fluorescence lifetime based pH measurement. To clarify, we have modified this paragraph on **Page 3**.

Question 2. A justification of the selected wavelengths will be helpful. Why is 600 nm chosen here? Is it the optimal wavelength providing the best sensitivity to the pH difference? Same to other two wavelengths used in the work.

Answer: The absorption at 565 nm presents the isosbestic point (i.e., the pH independent point). The isosbestic point can serve as the internal standard without need for a secondary reference probe. The 600 nm is picked because the optical absorption of SNARF-PAA NP at this wavelength has a large dynamic range when the pH changes from 6 to 8. The selection of this wavelength can lead to better sensitivity in pH measurement. This discussion has been added on **Page 5**. The newly added discussion about the selection of the 576 nm and 584 nm wavelengths can be found in the section “Method of quad-wavelength PA ratiometric imaging of pH” (**Page 14**).

Question 3. The authors may also want to add a simple explanation about the underline mechanism of pH sensing using SNARF-5F.

Answer: As suggested, the following material has been added in the Introduction section (**Page 4**).

“A well-known ratiometric pH indicator, SNARF-5F, i.e. 5-(and-6)-Carboxylic Acid, has been explored for qualitative pH measurement in vivo. The emission spectrum of SNARF-5F undergoes a pH-dependent wavelength shift, allowing the ratio of the fluorescence intensities at two emission wavelengths to be used for measurement of pH. Similar to the emission spectrum, the absorption spectrum of SNARF-5F is also a function of pH (as shown in Fig. S1 in Supplementary Materials). Therefore, following the idea of fluorescent dual-wavelength ratiometric measurement, PA absorption dual-wavelength ratiometric measurement of SNARF-5F has been explored, and its capability in detecting 0.1 pH changes has been demonstrated”.

Question 4. Line 142, the authors may want to justify the linear fitting in the ratiometric measurement. It seems to me that a higher order fitting is more accurate. The same question is for Fig. 4a.

Answer: We agree with the reviewer that a higher order fitting may turn out to be more accurate. However, a higher order of fitting will bring complication to the computation. Therefore, in this study, we applied the linear fitting, which has led to satisfactory accuracy in pH measurement. In the future, higher order fitting methods will be tried, and their potential improvement in accuracy will be explored.

Question 5. Please quantify the measurement errors (or accuracy) in Fig. 4b.

Answer: To quantify the measurement errors in **Fig. 4b**, the mean and the standard deviation of the pH levels in each PA image were calculated, and have been included in the updated **Fig. 4b**. The measurement accuracy was better than 0.1 pH.

Question 6. Line 194, what is the standard deviation of the measurements *in vivo*? Are the difference between the two measurements larger than the measurement accuracy quantified in Fig. 4b?

Answer: The standard deviations of the pH measurements in the tumor *in vivo* have now been added (6.73 ± 0.29 in the inner area of the tumor and 6.97 ± 0.35 in the outer area of the tumor) (see **Page 8**). The difference between the means from the two areas (inner and outer) is 0.24 pH which is larger than the accuracy of 0.1 pH as quantified in **Fig. 4b**, supporting our claim that the inner core of the tumor was more acidic.

Question 7. Fig. 5, how is the ROI of the tumor region selected?

Answer: Each PA functional image is super-imposed on the gray-scale US image acquired at the same time using the same acquisition system. Naturally co-registered with the PA image, the US image works well in delineating the tissue structures including the tumor boundary. This additional detail has been added in the section of “*In vivo* quad-wavelength ratiometric PAI of tumor” on **Page 8**.

Question 8. Fig. 5, please discuss if the measurement accuracy would decrease with the penetration depth?

Answer: Please see our answer to Question #3 in “Major Concerns”.

Question 9. Line 297, if a pixel-by-pixel ratio is calculated, will some pixels with weak signals give large errors due to noise? i.e., it is always a dangerous operation to divide two small numbers? Did the authors performed any thresholding before calculating the ratio?

Answer: We agree that, when pixels have weak PA signals, calculating the ratios among different wavelengths, as performed in quad-wavelength PA ratiometric imaging, could have larger errors caused by the background noise. However, this is not a concern for the study on phantoms, because the PA measurements of the phantoms, without being covered by optically scattering tissues, had excellent signal to noise ratio. This indeed could be a problem for imaging of mouse tumors *in vivo*. To reduce the possible errors caused by the background noise, a universal threshold of 6 dB above the background noise level was set for PA intensity images, meaning that the pixel-by-pixel calculation of the pH level was performed only for those with intensities above the 6 dB threshold. This additional detail has been added in the section of “PAI of tumor pH in a mouse model *in vivo*” (**Page 18**).

Question 10. Fig. 6c, why is the particle distribution so inhomogenous if there is no tumor?

Answer: Since the distribution of the locally injected NPs in the thigh is relying on perfusion, instead of a single injection of a high dose, multiple spatially scattered injections, each with a smaller dose, were applied aiming at achieving a more homogenous distribution of NPs in the thigh. This is the main reason why **Fig. 6c** from the normal tissue also shows some inhomogeneity. This additional detail has been added in the “PAI of tumor pH in a mouse model *in vivo*” section (**Page 18**).

Question 11. *The authors may consider citing the previous work using SNARF-5F as PAI contrast for pH sensing, by Chatni et al.. Journal of Biomedical Optics 16, 100503*

Answer: This reference has been added (**Reference #25**).

Question 12. *It would be helpful if the authors can add more details of their PAI system, such as ultrasound frequency, optical fluence, and image reconstruction method.*

Answer: A new section of “Imaging system” has now been added in the “Materials and Methods” (**Page 13**).

Question 13. *Fig. 6d, is the pH value in the tumor significantly different from that in the control?*

Answer: A t-test was performed with a hypothesis that the pH quantified by the quad-wavelength PA ratiometric imaging cannot differentiate the tumors and the normal thigh muscles. A $P < 0.001$ was reached, suggesting that the pH values in the tumors measured by PA imaging were significantly different from those in the controls. This paragraph on **Page 9** has been rephrased to make the claim clear.

REVIEWER #2:

In the paper entitled “In vivo quantitative imaging of tumor pH: Nanosonophore assisted multi spectral photoacoustic imaging”, the authors introduce a technique to quantitatively measure tumor pH in vivo using tumor-targeted pH sensing nano-probes and an imaging modality that combines photoacoustics and ultrasound. The authors used pH sensitive nanosonophores, which are encapsulated optical pH indicator (SNARF-5F) in polyacrylamide nanoparticles, and added surface modification for both tumor targeting and avoidance of the immune system. The authors used different wavelengths in PA imaging to separate the contrast agents signal from oxy and deoxy hemoglobin, and thereby calculate the pH in tumor environment. The method was demonstrated both in vitro and in vivo.

Overall, the results in the paper are adequately presented. However, I think that for Nature Communication, the novelty of the paper is very marginal and not sufficient. In particular, compared to the group’s previous published work in this field (Ray et al., 2013, Analyst, “Sonophoric nanoprobe aided pH measurement in vivo using photoacoustic spectroscopy), which reports both in vitro and in vivo pH measurements, including “successful pH sensing in vivo on a rat joint model, with a precision better than 0.1 pH units” and the work on tumor targeted nanoparticles (Ray et al., 2011, Nano Res., “Targeted blue nanoparticles as photoacoustic contrast agent for brain tumor delineation”). The modification the group performed to their previous work is the use of four wavelengths rather than two to extract the concentration of the pH sensitive dye more accurately, and thereby quantify the pH. Extraction of dyes in whole blood using multiple wavelengths has already been reported (e.g., Li et al., 2008, Proceedings of IEEE 96, “Simultaneous molecular and hypoxia imaging of brain tumors in vivo using spectroscopic photoacoustic tomography”).

Answer: The main innovation of this work is reflected by the achievement of a successful truly quantitative imaging of tumor pH *in vivo*, which is made possible by the proposed imaging technology combining quad-wavelength PAI and SNARF-PAA NPs. All of the previous studies, including ours at Michigan, only validated the feasibility of multi-wavelength PAI for a qualitative measurement of pH levels. The precision of better than 0.1 pH reported in our paper published in the *Analyst* was a demonstration of “sensitivity” rather than “accuracy” (i.e., 0.1 pH change can be sensed, but the absolute

pH value cannot be quantified). As discussed in the manuscript, the major challenge in achieving a real quantitative pH imaging is the background optical absorption from other tissues (mainly oxy- and deoxy-hemoglobin), which is successfully solved by the method described in this manuscript. As another beauty of this method, besides the pH level, another two important functional parameters reflecting the tumor microenvironment, including blood volume and blood oxygenation, can also be quantitatively mapped at the same time. The Introduction and the Discussion sections have been largely revised to emphasize the innovation of this work.

We also want to point out that the proposed quad-wavelength ratiometric PAI has many technical difficulties when performed in the complicated *in vivo* environment. One difficulty is the potential error induced by the wavelength-dependent optical attenuation in tissue. To solve this challenge, we intentionally selected the four optical wavelengths to be close so that the optical spectral range for quad-wavelength PA ratiometric imaging is relatively small. In other words, the optical spectrum selected can differentiate the optical spectra of HbO₂, Hb and pH-dependent SNARF-PAA NPs; while the optical attenuation of the selected wavelengths in tissue can be significantly different. We have further studied the potential error in the pH quantification due to the spectroscopic optical attenuation in tissue, as shown in the newly added **Fig. S8** in Supplementary Materials. This study demonstrates that the quartet of 565, 576, 584, and 600 nm form a good set of wavelengths for quad-wavelength PAI of pH. Another difficulty in performing the quad-wavelength PAI is the imaging speed. Powered by a GPU card, our dual-modality system can acquire PA and US images from the same sample at the same time, both in real-time fashion, with a frame rate of 10 Hz. The fast imaging speed is highly valuable for functional PA imaging, because multi-wavelength images need to be acquired within a relatively short time period before functional parameters potentially change. Discussions about these difficulties and technical developments have been added in the updated manuscript (see **Page 16-17** and **Page 13**). After all these technical developments, a truly quantitative PA imaging of pH as a key functional parameter of the tumor microenvironment has been made possible for the first time.

Other questions:

Question 1. *The authors suggest that the imaging technology reported may be used to monitor “cancer microenvironment in vivo, WITH IMPORTANT RAMIFICATION FOR THE OPTIMAL CHOICE OF CHEMOTHERAPY” – However, the results don’t support such a claim.*

Answer: Since tumor pH level is highly relevant to chemotherapy, we expect that a novel pH imaging technology could contribute to optimizing chemotherapy. The following discussion has been added in the Discussion section (**Page 11**). “It has been reported that the cytotoxicities of some drugs, such as daunorubicin, doxorubicin, and mitoxantrone, are reduced under acidic condition; while other drugs such as chlorambucil, cyclophosphamide, and 5-fluorouracil show higher cytotoxicity at lower pH. Therefore, being capable of quantifying the pH level of the tumor microenvironment using an imaging technology may enable optimization of chemotherapy and facilitate personalized treatment for cancer patients.”

Question 2. *In figure 3, the authors show how the total hemoglobin concentration as well as the oxygen saturation affects the ratio-metric method for calculating the pH. I think that these results are obvious and expected considering that when one would like to extract three parameters (HbO₂, Hb, and pH) at least three independent equations are required, and therefore the use of two wavelengths is not sufficient. In other words, the authors should explain why the dual-wavelength ratiometric method was originally suggested to work in whole blood environment, in which the oxygen saturation is unknown.*

Answer: In previous studies, including ours, a dual-wavelength ratiometric method was used by following the established method of dual-wavelength fluorescent ratiometric measurement. Similar to its emission spectrum which undergoes a pH-dependent wavelength shift, the absorption spectrum of

SNARF-5 is also a function of pH. The purpose of previous initial studies was to demonstrate that a dual-wavelength PA measurement can also sense the pH change in biological samples. Without worrying about background optical absorption, the dual-wavelength PA imaging aiming at qualitative pH measurement has less technical challenges. However, when the two forms of hemoglobin also contribute to PA signals at different wavelengths, providing accurate spatial pH information is nearly impossible without careful consideration of the background optical absorption spectrum. Additional discussion about the finding and limitation of previous studies based on dual-wavelength ratiometric imaging has been added in the Introduction section (**Page 4**).

Question 3. *The phantom experiments of the quad-wavelength ratiometric method were done with 1% of whole blood (Line 177). It seems to me that 1% of whole blood diluted in 99% of buffer is not sufficient to prove that the method can efficiently separate the pH from whole blood in vivo. Moreover, the main assumption in using the linear equations to extract the concentrations of different absorbers is linearity. It was previously shown that spectroscopic PA measurements are affected by nonlinearity of the blood absorption (e.g., Danielli et al., 2015, APL, "Nonlinear photoacoustic spectroscopy of hemoglobin"). Have the authors verified that in whole blood the PA signal is linear with incident power and that the extraction of the dye is accurate?*

Answer: The blood content in tumor varies, depending on the tumor model, tumor size, and tumor aggressiveness. As reported in the literature [AJNR. American journal of neuroradiology 29(4), 694-700 (2008)], the blood content in human glioma tumor ranges from 0.95 to 2.79%. Therefore, adding 1% blood in each phantom is reasonable, and has enabled a proof-of-principle experiment to demonstrate the better performance of quad-wavelength PAI over dual-wavelength PAI. Moreover, in later simulations, the error in dual-wavelength ratiometric PA measurement of pH as a function of the blood content was more extensively and quantitatively studied, as the result shown in **Fig. 3d**. In the revised manuscript, the rationale for 1% blood content in phantoms has been added to as section of "PAI on phantoms" (**Page 17**).

Question 4. *In Methods, The PA system is not detailed. What is the laser fluence at the surface, what is the resolution and imaging depth of the system?*

Answer: A new section "Imaging system" has been added (**Page 13**) to introduce the details of the PA and US dual-modality system.

REVIEWER #3:

The manuscript describes a method of measuring pH in the tumor non-invasively. Although the approach of using photoacoustics for pH measurements is not quite unique, the results are promising. Overall, the manuscript requires some additional input from the authors as specified below.

Question 1. *The authors used a long-standing and commercially available ratiometric pH indicator SNARF. One of the reasons that this dye has not been used in vivo is due to strong attenuation by both scattering and tissue absorption. The general approach of using red dyes for imaging does not go beyond the small animal models with subcutaneous tumors. Even this model requires relatively large amount of the probe as the authors used (250 mg NP, which is an equivalent of 17 g of human injection). Please comment. Can the authors provide an estimate of how far the signal can penetrate at 560-600 nm?*

Answer: To address this comment, the following paragraph has been added in the Discussion section (Page 10). To further evaluate penetration and its effect on pH measurement accuracy, additional experimental results have been included in the Supplementary Materials (Fig. S8 and Fig. S9).

“The commercially available SNARF-5F is considered as a long-wavelength fluorescent pH indicator, and has been well-developed and widely used for fluorescence based measurement of pH in vitro. This is the reason that the SNARF-5F was employed to build the pH sensing PAA NPs in this proof-of-principle study. Working in the spectral range of 565-600 nm, quantitative imaging of pH by quad-wavelength ratiometric PA measurement can be achieved with satisfactory accuracy in subsurface tissue at a depth up to 6 mm. We have further estimated the error in pH quantification due to the optical attenuation at different wavelengths for quad-wavelength PA ratiometric imaging, as described in Supplementary Materials (Fig. S8). As expected, the error becomes larger when the imaging depth increases. However, at 6-mm depth in optically scattering tissue, the error caused by the optical attenuation was less than 0.16 pH. This error could be further reduced by compensating the wavelength-dependent optical attenuation when simulating the point-by-point pH levels. In another experiment, as described in the Supplementary Materials (Fig. S9), we have further assessed the sensitivity limit of PA imaging in detecting the SNARF-PAA NPs in subsurface tissue. At the depth of 6 mm in optically scattering tissue, SNARF-PAA NP solution at very low concentration of 0.05 mg/ml can still be detected with a good signal-to-noise ratio (SNR) over 20 dB. The 6-mm depth achieved by PA pH imaging is one order of magnitude deeper than that accessible by fluorescence microscopy. With the current imaging depth, many clinical applications on relatively superficial tumors become possible, such as head and neck cancer, colorectal cancer, and cervical cancer, which are all associated with an acidic tumor microenvironment. For imaging of deeper tumors, pH indicating dyes that absorb at longer optical wavelengths (e.g. 650-950 nm) will be needed to further improve the optical penetration. One advantage of our PAA NP system is that replacing the dye inside the NP is relatively simple.”

Question 2. A large dosage might cause a rapid change in the metabolism that might lead to the change in the pH of the tissue. Please comment.

Answer: To the best of our knowledge, the metabolic change in the body caused by systemic injection could be rapid, but the induced change in tissue pH should be small, mainly due to the buffer systems in the mammal's body. In addition, we don't expect the injection of a neutral agent (~pH 7) to largely change the system pH level.

Question 3. How do authors see the application of the proposed method for the pH measurements? According to the provided results the pH inside the tumor is 6.7 which is within the range of normal pH level in the body. In that case and using the proposed method the location of the tumor might never been found. Please describe how the probes can be used for diagnostics or drug selection as mentioned in the discussion?

Answer: We agree with the reviewer that the main contribution of pH imaging to clinical management of cancer might be image-guided treatment instead of cancer diagnosis. It has been reported that the cytotoxicities of some drugs such as daunorubicin, doxorubicin, and mitoxantrone are reduced under acidic condition; while other drugs such as chlorambucil, cyclophosphamide, and 5-fluorouracil show higher cytotoxicity at lower pH. Therefore, being capable of quantifying the pH level of the tumor microenvironment using imaging technology may enable optimization of chemotherapy and facilitate personalized treatment for cancer patients. This additional discussion has been added to the Discussion section (Page 11), and the previous claim about diagnosis has been removed.

Question 4. *Did the authors perform any blocking studies to verify the uptake by the tumor? Does targeting with F3 work or that is another example of the EPR effect?*

Answer: We have done extensive *in vivo* studies in the past on the PAA NPs, and demonstrated that F3-peptide modified NPs work better than non-targeted NPs in terms of delivery efficiency. Example publications include Kopelman et al., *Neurosurgery*, **64**(5), 965-972. (2010); Kopelman et al., *Cancer Res.* **70**(21), 8674-83. (2010); and Kopelman et al., *Small*, **8**(6), 884-891. (2012). This discussion and the above references have been added in the “Synthesis SNARF-PAA NP and F3 surface modification” section (**Page 12**). In addition, in the section of “*In vivo* quad-wavelength ratiometric PAI of tumor” on **Page 8**, we revised the claim as “The NPs were preferentially accumulated in the tumor area gradually, which, we believe, is a combined result of the enhanced permeability and retention (EPR) effect and the tumor homing F3 peptides.”

Question 5. *Line 57 – 60. The manuscript only mentions MRI and PET. However, there is a large effort of developing pH optical probes primarily with NIR dyes that is not mentioned in the manuscript: Berezin et al Biophys. J, 2011, 20; 100(8):2063-72., Lee et al Photochem Photobiology, 2013, 89 (2) 326–331, Gilson et al Mol. Pharmaceutics, 2015, 12 (12), pp 4237–4246, Chen et al Advanced materials, 2015 6820-6827 and some others. Please provide more background on using optical probes for in vivo.*

Answer: Additional discussion about NIR pH dyes and the suggested references have been added in the Introduction section (**Page 3**).

Question 6. *Line 120-125. Often the dyes change their absorption and fluorescent properties in the presence of albumin that causes the shift of the isosbestic point due to a hydrophobic environment. Therefore ratiometric approach might still work but for a different set of wavelengths. Provide the abs spectra of the dye in the presence of albumin at different pH and comment.*

Answer: As suggested, the spectra of free SNARF-5F at different pH levels and their changes in the presence of human serum albumin have been added in the Supplementary Materials (**Fig. S1**).

By comparing **Fig. S1a** and **S1b**, we can see the changes caused by the interaction between SNARF-5F and albumin, including not only the shifts of the spectrum but also the changes in spectral shape. Moreover, the isosbestic point cannot be identified easily in **Fig. S1b**. We agree with the reviewer that, in case the pH-dependent optical absorption spectra of SNARF-5F in the presence of human serum albumin can be calibrated, ratiometric imaging of the pH level might still work but for a different set of wavelengths. However, the concentration of extracellular albumin in biological sample may vary, which can result in another uncertainty for pH measurement. All these potential problems can be solved by encapsulating SNARF-5F in PAA NPs, which keeps the albumin away from the indicator dye and, in addition, can also improve biocompatibility, reduce toxicity, and enable better cancer targeting (**Page 5**).

REVIEWER #4:

Overall description and comments:

The authors describe problems with existing approaches for quantitative in vivo imaging of pH and its importance for cancer diagnostics and monitoring a response to therapy. Their proposed solution is a smart nanoplatform that contains a standard pH-sensitive dye (SNARF 5F), which can be functionalized to target cancerous tissue. Their rationale is that the encapsulated dye is more stable than the free dye,

whose response changes in the presence of tissue protein/albumin. Instead of fluorescent imaging of the dye, however, the authors employ photoacoustic imaging with spectral unmixing to map total hemoglobin and tissue pH at depth and with good resolution (<200 μm). Through careful and methodical testing in photoacoustic phantoms, they develop calibration curves for sensing pH that are independent of the concentration of the dye or effects of hemoglobin/sO₂, which is often a confounding factor during imaging of tumors. Finally, they demonstrate proof-of-concept of in vivo pH imaging in tumors on the back of 4 mice using a 9L rat glioma cell line. The thigh was used as a control. They also describe clearance of the agents and give support that the nanoplatform is biocompatible. The study is quite novel and offers the cancer research community a new type of contrast agents that others can employ to map pH noninvasively for their small animal research. As long as the investigator has access to photoacoustic imaging and spectroscopy at visible wavelengths, they should be able to implement the agent and algorithms into their research. Although the penetration at visible wavelengths of a few millimeters is limited due to high scattering and absorption, this might be sufficient for orthotropic tumor imaging in mice. As a new noninvasive tool for imaging and sensing, the techniques could play an important role in developing new diagnostic techniques and therapies for cancer. The manuscript is well written with rationale arguments, provides sufficient background, and details the methodology in comparison with standard techniques. The methods section clearly describes the unmixing algorithm to separate background absorption (Due to hemoglobin) from the pH signal. Because this is a proof-of-concept study, it is acceptable that the authors have a relatively low # of animal subjects (4 mice) and trials for averaging across experiments to determine the calibrations (in some cases as few as an n=3). This was fine as the initial study in phantoms and mice. However, there are still some areas that need improvement, which are described below.

Major Comments:

Question 1: *There is very little discussion on the limitations of using the nanoagents for photoacoustic imaging and pH sensing at visible wavelengths. What is the penetration limit due to scattering and absorption? How does the accuracy change with depth in a highly scattering/absorbing environment? What about effects from background absorbers in tissue other than hemoglobin? What was the degree of photobleaching during the experiments (e.g., loss of PA signal per pulse normalized to fluence)?*

Answer: To evaluate the imaging depth of this technology and the accuracy change with depth in biological tissue, the result from an additional experiment has been included in the Supplementary Materials (**Fig. S8**). In addition, to evaluate the sensitivity of the imaging technology in detecting very low concentration of SNARF-PAA NPs in subsurface tissue (another way of studying the penetration limit), an additional experimental result has been included in the Supplementary Materials (**Fig. S9**).

In this study, we assumed that the major chromophores in the target biological samples are the two forms of hemoglobin, which is true in the spectral region of 565-600 nm. In the future, when other chromophores may be present in the target tissue, imaging at additional wavelengths will be needed. This discussion has been added in the Discussion section in **Page 10**.

To examine the photobleaching, the result from an additional experiment has been included in the Supplementary Materials (**Fig. S5**), demonstrating that our SNARF-PAA NPs are excellent in terms of photostability.

Question 2: *It appears that the phantom work was done at room temperature. Did the investigators look at the effects of temperature (e.g., body temperature) and whether that affects the calibration curve in a significant or measurable way?*

Answer: To study the temperature stability of the SNARF-PAA NPs, we have generated calibration curves at different temperatures, as shown in the newly included **Fig. S6** in the Supplementary Materials. We do not notice a significant difference between the curves at room temperature and body temperature.

Question 3: *Fig 2a employed a different setup (e.g., V312) than the Fig 2b and the other experiments (Verasonics). Was there a reason that the Verasonics system could not be used for all of the experiments for a fair comparison?*

Answer: **Fig. 2a** and **Fig. 2b** were both measurements from a UV-VIS spectrometer. We assume that the reviewer actually refers to **Fig. 4a** and **Fig. 4b** which were acquired using a single-element ultrasonic transducer and the Verasonics imaging system respectively. When fast imaging of a sample is not the purpose (as for **Fig. 4a**), using the Verasonics system is not necessary. Instead, PA measurement based on a well-calibrated single-element transducer can provide more accurate calibration curves for later imaging experiments on phantoms and animals.

Question 4: *Was a breathing artifact apparent during mouse imaging? How was this handled as a source of error during spectroscopy/unmixing.*

Answer: Equipped by the Verasonics system, high-speed PA and US dual-modality B-scan imaging of the same tumor can be achieved (frame rate: 10 Hz). To improve the signal-to-noise ratio, the PA images acquired at each wavelength were averaged 50 times. Considering that the wavelength switching time was 5-10 seconds, the total time period for PA imaging at the four wavelengths was less than 60 seconds. To avoid potential motion artifacts, the animal, especially the tumor area under the scan, was fixed tightly during image acquisition, using a home-fabricated fixation device. This additional detailed information has been added in the “PAI of tumor pH in a mouse model in vivo” section (**Page 17**).

Question 5: *Can the authors add a bit more detail on how the light was delivered to the mouse? Was it through a dual fiber bundle? What was the energy out of the laser and fluence on the skin? How much averaging was performed (reps per wavelength)? What was the wavelength switching time?*

Answer: The information about light delivery and light fluence applied on the skin are included in the newly added section entitled “Imaging system” in “Materials and Methods” (**Page 13**). Also see our answer to Question 4 in the above.

Question 6: *Was 3D imaging performed (or possible) with your setup?*

Answer: 3D imaging of pH is possible, and can be more valuable in evaluating the tumor microenvironment. However, using the current system, the data acquisition time would be very long. Therefore, in this proof-of-principle study, only 2D imaging was conducted. Some additional discussion about 3D imaging has been added in the Discussion section (**Page 10**).

Question 7: *Are there any potential human applications of the technology given the limited penetration of a few millimeters? What are the current challenges/limitations that need to be overcome?*

Answer: The following discussion about human application and imaging depth has been added in the Discussion section (**Page 11**). Besides imaging depth, another challenge/limitation is the imaging speed which has also been further discussed (**Page 10**).

“The 6-mm depth achieved by PA pH imaging is one order of magnitude deeper than that accessible by fluorescence microscopy. With the current imaging depth, many clinical applications on relatively superficial tumors become possible, such as head and neck cancer, colorectal cancer, and cervical cancer which are all associated with acidic tumor microenvironment. For imaging of deeper tumors, pH indicating dyes that absorb at longer optical wavelengths (e.g. 650-950 nm) will be needed to further improve the optical penetration. One advantage of our PAA NP system is that replacing the dye inside the NP is relatively simple.”

Minor Comments:

1) Figure caption/legend for Figure 3 (page after references): Title does not seem to relate to the plots/subplots in this figure. There are no “mechanisms” described in Figure 3. Please restate.

Answer: As suggested, the caption for **Figure 3** has been updated.

2) Page 1 (intro). It is stated that MR and CT require contrast agents for pH mapping. CEST MRI does not necessarily require contrast agents, since it measures changes in water saturation.

Answer: This paragraph in the Introduction talking about the limitations of MR and PET based technologies has been revised (**Page 3**).

3) Paragraph from Lines 70-83 is poorly written with many grammatical errors.

Answer: As suggested, the entire paragraph has been revised.

4) A few times (e.g., see Fig. 2 caption) “HSA” is misspelled “HAS” Please do a “search and replace.”

Answer: Corrected.

Reviewers' comments:**Reviewer #1 (Remarks to the Author):**

I am happy with the revised manuscript. All of my concerns and questions have been adequately addressed by the authors, especially the additional materials about the wavelength-dependent optical attenuation. I highly recommend the publication of this beautiful work in the prestigious Nature Communications.

Reviewer #2 (Remarks to the Author):

The manuscript by J Jo et al., entitled "In vivo quantitative imaging of tumor pH: Nanosonophore assisted multi spectral photoacoustic imaging", has been thoroughly revised with more details and several new results.

In their revised manuscript, the authors did a good job clarifying the novelty of their work in the context of previous research in the field, including their own. The authors replied to the points raised by the reviewers. Therefore, the manuscript can be accepted for publication in Nature Communication pending on a couple of minor comments.

1. Some parameters of the optical system are still missing (e.g., the beam size, the field of view)
2. Fig 5c present the sO₂ in the tumor area 75 minutes after the injection. Can the authors verify their quad-wavelength ratiometric PAI system by demonstrating that the sO₂ in the tumor area at time 0 is similar to time 75 (If indeed the NP only contribute to the measurement of the pH and not the sO₂)?

It would be interesting to see whether the injection of the NP affects the sO₂ of the tumor area.

Reviewer #3 (Remarks to the Author):

I believe the authors responded to all critiques and substantially revised the manuscript

Reviewer #4 (Remarks to the Author):

While the authors have adequately responded to most of the previous concerns, I still have a couple of questions/recommendations.

Question 3: Fig 2a employed a different setup (e.g., V312) than the Fig 2b and the other experiments (Verasonics). Was there a reason that the Verasonics system could not be used for all of the experiments for a fair comparison?

Answer: Fig. 2a and Fig. 2b were both measurements from a UV-VIS spectrometer. We assume that the reviewer actually refers to Fig. 4a and Fig. 4b which were acquired using a single-element ultrasonic transducer and the Verasonics imaging system respectively. When fast imaging of a sample is not the purpose (as for Fig. 4a), using the Verasonics system is not necessary. Instead, PA measurement based on a well-calibrated single-element transducer can provide more accurate calibration curves for later imaging experiments on phantoms and animals.

1) Yes, I was referring to Fig. 4a and 4b. Not sure I understand the logic. Why would a “well-calibrated” focused single element transducer provide more accurate results (for calibration) than a “well calibrated” Verasonics array and imaging system? Since the Verasonics system was employed for the mouse experiments, wouldn't it make sense to examine the calibration curves for that system, as well (or at least compare it with the single element transducer)? The Verasonics system enables imaging of the samples (rather than just a point measurement) and its high speed would still allow for a similar averaging profile per unit time.

2) Regarding Figure S5 on photostability, since photobleaching depends on the laser repetition rate *and* fluence, photobleaching for a pulsed laser is typically reported as "cumulative dose" in J/cm^2 (or equivalent) as standard units rather than "illumination time."

Responses to Reviewers' Comments

We greatly appreciate the additional comments from the reviewers. Below are our responses.

Reviewer #1 (Remarks to the Author):

I am happy with the revised manuscript. All of my concerns and questions have been adequately addressed by the authors, especially the additional materials about the wavelength-dependent optical attenuation. I highly recommend the publication of this beautiful work in the prestigious Nature Communications.

Reviewer #2 (Remarks to the Author):

The manuscript by J Jo et al., entitled "In vivo quantitative imaging of tumor pH: Nanosonophore assisted multi spectral photoacoustic imaging", has been thoroughly revised with more details and several new results.

In their revised manuscript, the authors did a good job clarifying the novelty of their work in the context of previous research in the field, including their own. The authors replied to the points raised by the reviewers. Therefore, the manuscript can be accepted for publication in Nature Communication pending on a couple of minor comments.

Q 1. *Some parameters of the optical system are still missing (e.g., the beam size, the field of view)*

Answer: The light beam on the sample surface formed a rectangle shape with a size of 1.5 cm by 3 cm. This information has been added to the "Imaging system" in Materials and Methods (page 13).

Q 2. *Fig 5c present the sO₂ in the tumor area 75 minutes after the injection. Can the authors verify their quad-wavelength ratiometric PAI system by demonstrating that the sO₂ in the tumor area at time 0 is similar to time 75 (If indeed the NP only contribute to the measurement of the pH and not the sO₂)? It would be interesting to see whether the injection of the NP affects the sO₂ of the tumor area.*

Answer: The sO₂ in the tumor area at 0 minute (before injection) was added in the Supplementary Materials (**Supplementary Fig. 10**). This image at 0 min shows similar sO₂ level as the image acquired at 75 minutes (**Fig. 5c**). The center area of the tumor also has relatively low oxygen saturation. Thus the injection of the NPs for pH imaging did not strongly affected the functional sO₂ imaging result.

Reviewer #3 (Remarks to the Author):

I believe the authors responded to all critiques and substantially revised the manuscript

Reviewer #4 (Remarks to the Author):

While the authors have adequately responded to most of the previous concerns, I still have a couple of questions/recommendations.

Fig 2a employed a different setup (e.g., V312) than the Fig 2b and the other experiments (Verasonics). Was there a reason that the Verasonics system could not be used for all of the experiments for a fair comparison?

Answer: Fig. 2a and Fig. 2b were both measurements from a UV-VIS spectrometer. We assume that the reviewer actually refers to Fig. 4a and Fig. 4b which were acquired using a single-element ultrasonic transducer and the Verasonics imaging system respectively. When fast imaging of a sample is not the purpose (as for Fig. 4a), using the Verasonics system is not necessary. Instead, PA measurement based on a well-calibrated single-element transducer can provide more accurate calibration curves for later imaging experiments on phantoms and animals.

1) Yes, I was referring to Fig. 4a and 4b. Not sure I understand the logic. Why would a “well calibrated” focused single element transducer provide more accurate results (for calibration) than a “well calibrated” Verasonics array and imaging system? Since the Verasonics system was employed for the mouse experiments, wouldn't it make sense to examine the calibration curves for that system, as well (or at least compare it with the single element transducer)? The Verasonics system enables imaging of the samples (rather than just a point measurement) and its high speed would still allow for a similar averaging profile per unit time.

Answer: Following the suggestion from the reviewer, we have also measured the calibration lines using the Verasonics system, as shown by the newly added **Supplementary Fig. 11** in Supplementary Materials. As we can see, the results from the Verasonics system are close to those from the single-element transducer, demonstrating that the calibration lines are independent of the detection system.

2) Regarding Figure S5 on photostability, since photobleaching depends on the laser repetition rate and fluence, photobleaching for a pulsed laser is typically reported as "cumulative dose" in J/cm² (or equivalent) as standard units rather than "illumination time."

Answer: Following this suggestion, we have now updated the label of the x-axis in **Supplementary Fig. 5** with “Cumulative light dose (J/cm²)”.